# Deep invariant networks with differentiable augmentation layers

**Cédric Rommel, Thomas Moreau & Alexandre Gramfort**
Université Paris-Saclay, Inria, CEA, Palaiseau, 91120, France
`{firstname.lastname}@inria.fr`

## Abstract

Designing learning systems which are invariant to certain data transformations is critical in machine learning. Practitioners can typically enforce a desired invariance on the trained model through the choice of a network architecture, e.g. using convolutions for translations, or using data augmentation. Yet, enforcing true invariance in the network can be difficult, and data invariances are not always known *a piori*. State-of-the-art methods for learning data augmentation policies require held-out data and are based on bilevel optimization problems, which are complex to solve and often computationally demanding. In this work we investigate new ways of learning invariances only from the training data. Using learnable augmentation layers built directly in the network, we demonstrate that our method is very versatile. It can incorporate any type of differentiable augmentation and be applied to a broad class of learning problems beyond computer vision. We provide empirical evidence showing that our approach is easier and faster to train than modern automatic data augmentation techniques based on bilevel optimization, while achieving comparable results. Experiments show that while the invariances transferred to a model through automatic data augmentation are limited by the model expressivity, the invariance yielded by our approach is insensitive to it by design.

## 1   Introduction

Inductive biases encoding known data symmetries are key to make deep learning models generalize in high-dimensional settings such as computer vision, speech processing and computational neuroscience, just to name a few. Convolutional layers [1] are the perfect illustration of this, as their translation equivariant property allowed to reduce dramatically the size of the hypothesis space compared to fully-connected layers, opening the way for modern computer vision achievements [2]. This illustrates one way to encode desired symmetries in deep learning models by hard-coding them directly in the network architecture. An alternative way is to use data augmentation, where transformations encoding the desired symmetries are applied to the training examples, thus adding additional cost when such symmetry is not recognized by the model. While in the first case invariances are *built in* the network by design and are therefore a hard constraint, data augmentation promotes certain invariances more softly. As opposed to being *built-in*, desired invariances are here *trained-in* [3].

In both cases, the invariances present in the data are not always known beforehand. While relevant invariances are intuitive for some tasks such as object recognition (e.g. a slightly tilted or horizontally flipped picture of a mug, still represents a mug), the same cannot be said for many important predictive tasks such as classifying brain signals into different sleep stages [4, 5]. In order to be able to tackle this problem of learning optimal systems from complex data two strategies are pursued in the literature: neural architecture search (NAS) which aims to find the best architectural elements from the training data, and automatic data augmentation (ADA) which aims to learn augmentation policies automatically from a given dataset. Both fields tackle the problem very similarly, by parametrizing the

36th Conference on Neural Information Processing Systems (NeurIPS 2022).

network architecture or the augmentation policies, leading to a bilevel hyperparameters optimization problem [6]. While these techniques allowed to find architectures and augmentations capable of outperforming the state-of-the-art in some cases, solving such bilevel optimization problems is often difficult and computationally demanding [7, 8, 9].

This work investigates how to learn data invariances directly from the training data, avoiding the complex bilevel structure of ADA. To this end, we propose to integrate learnable data augmentation layers within the network, and train them together with other model parameters using a novel invariance promoting regularizer. Our approach extends previous works into a very general framework that goes beyond computer vision tasks, as it can incorporate any type of differentiable augmentations. We demonstrate the *versatility* of our method on two well controlled simulated settings, as well as an image recognition and a sleep stage classification dataset. We show that our approach can correctly select the transformations to which the data is invariant, and learn the true range of invariance, even for nonlinear operations. Our experiments also demonstrate that the data augmentation layer proposed here leads to almost perfect built-in invariances irrespective to the complexity of the original network it is added to. Moreover, we are able to achieve comparable performance and speed as state-of-the-art ADA approaches on our sleep staging experiment with a completely *end-to-end* approach, avoiding tedious bilevel optimization parameters. The accompanying code can be found at https://github.com/cedricrommel/augnet.

## 2   Related work

**Automatic data augmentation**   Automatic data augmentation aims to learn relevant data invariances which increase generalization power. More precisely, ADA is about searching augmentations that, when applied during the model training, will minimize its validation loss. This objective is summarized in the following bilevel optimization problem:

$$
\begin{aligned}
\min_{\mathcal{T}} \quad & \mathcal{L}(\theta^*|D_{\text{valid}}) \\
\text{s.t.} \quad & \theta^* \in \arg\min_{\theta} \mathcal{L}(\theta|\mathcal{T}(D_{\text{train}})) \ ,
\end{aligned}
\tag{1}
$$

where $\mathcal{T}$ is an augmentation policy, $\theta$ denotes the parameters of some predictive model, and $\mathcal{L}(\theta|D)$ its loss over the set $D$. Initial ADA approaches such as AutoAugment [6] and PBA [10] use discrete search algorithms to approximately solve (1). Despite the impressive results obtained, they are tremendously costly in computation time, which makes them impractical in many realistic settings. In an attempt to alleviate this limitation, Fast AutoAugment [11] proposes to solve a surrogate density matching problem, which breaks the bilevel structure of (1). It is hence substantially faster to solve, since it does not require to train the model multiple times. However, this method needs a pre-trained model and its success highly depend on whether the latter was able to learn relevant data invariances on its own. Another way of carrying ADA efficiently is by using gradient-based algorithms, as proposed in Faster AutoAugment [12], DADA [13] and ADDA [14]. While Faster AutoAugment also tackles a surrogate density matching problem, DADA and ADDA solve the bilevel problem (1) directly. These ADA methods are the most related to our work since we rely on the same differentiable relaxations of standard augmentation transformations. However, we are mostly interested in building the learned invariances into the model, which is out of the scope of these methods. Indeed, they require substantially more overhead than our approach since they learn augmentations from the validation set and need to retrain the model after the augmentation search is completed. Moreover, DADA and ADDA are based on an alternating optimization of the inner and outer problems [15], which suffer from noisy hypergradients on the outer level due to the stochastic inner problem, and require careful tuning of the outer learning rate [7, 8, 9].

**Embedding invariances within neural network architecture**   A vast literature has focused on encoding predefined invariances or equivariances into neural network architectures. For instance, group convolutions allow to extend traditional convolutional layers to groups of affine transformations other than translations [16]. More related to our work, DeepSets [17] encode permutation invariance by summing networks predictions. Although related, these methods are not designed for learning symmetries from the data, which is the objective of this study.

Prior work on this matter from [18] proposes to learn invariances using the marginal likelihood in the context of Gaussian processes. In contrast, we are mostly focused on deep neural networks. [3]

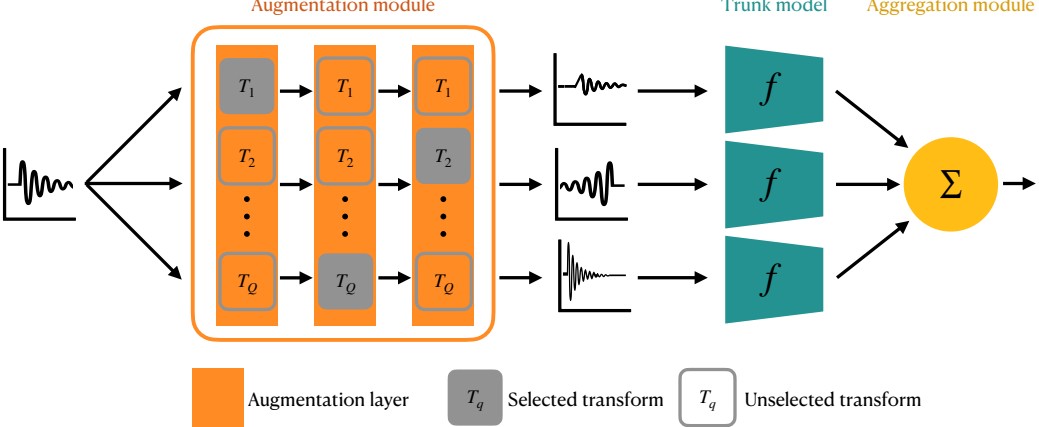

Figure 1: General architecture of AugNet. Input data is copied $C$ times and randomly augmented by the augmentation layers forming the *augmentation module*. Each copy is then mapped by the *trunk model $f$*, whose predictions are averaged by the *aggregation module*. Parameters of both $f$ and the augmentation layers are learned together from the training set.

suggests to discover data symmetries and build them into neural networks by learning weight sharing patterns in a meta-learning framework. Most related to our work is Augerino [19], which allows to learn invariances to affine transformations from the training data, by adding a learnable sampling brick to an existing model and averaging its predictions. They propose to learn the range of distributions from which affine transformations are sampled, by parametrizing the corresponding Lie group in terms of its Lie algebra. As it is unclear how to extend Augerino beyond Lie groups such as affine transformations, its scope of application is rather limited, making it mostly taylored for computer vision tasks. In this work we build on their ideas, extending Augerino to more diverse applications with hierarchical differentiable data augmentation layers built into the networks. Moreover, in addition to learning sampling distribution ranges, we also learn to *select* transformations which encode the strongest data invariance.

## 3 AugNet: A general framework to embed data augmentation into neural networks

### 3.1 Preliminaries

Let us consider a dataset $(x_1, y_1), \ldots, (x_N, y_N)$ of observations sampled from an unknown distribution $\mathbb{P}_{X,Y}$ over $\mathcal{X} \times \mathcal{Y}$. In a supervised setting, one wants to use this data to train a model $f : \mathcal{X} \to \mathcal{Y}$ (e.g. a neural network) so that it can predict $y$ from new observations $x$ sampled from the marginal $\mathbb{P}_X$. Now, suppose that the data joint distribution $\mathbb{P}_{X,Y}$ is invariant to a certain group $G$ of transformations acting on $\mathcal{X}$, i.e. for any $g \in G$ the distribution $\mathbb{P}_{X,Y}$ is close (in some sense) to the transformed distribution $\mathbb{P}_{gX,Y}$ (*cf.* [20] for further details). This means for example that, if $Y = y$ is the class "dog", than the probability of sampling an image $X = x$ of a dog is close to the probability of sampling the transformed image $gx$. In this situation, one would like the model $f$ to have this same invariance by design, so that it does not have to learn it from scratch and can better generalize to new observations of $\mathbb{P}_{X,Y}$. In other words, we would like $f(gx) = f(x)$ for any $g \in G$. One way of achieving this is to average $f$ over $G$ endowed with a uniform distribution $\nu_G$:

**Proposition 3.1.** *For a given model $f : \mathcal{X} \to \mathcal{Y}$ and a compact group of transformations $G$ acting on $\mathcal{X}$, $\bar{f} : \mathcal{X} \to \mathcal{Y}$ defined as*

$$\bar{f}(x) = \mathbb{E}_{g \sim \nu_G}\left[f(gx)\right] \ ,$$

*where $\nu_G$ is a uniform distribution on $G$, is invariant through the action of $G$.*

The proof follows from the assumption that $\nu_G$ is uniform and $G$ is a group, hence stable by composition and inverse. So for any $u \in G$, $gu^{-1}$ is also in $G$ and

$$\bar{f}(ux) = \mathbb{E}_{g \sim \nu_G}\left[f(gux)\right] = \mathbb{E}_{hu^{-1} \sim \nu_G}\left[f(hx)\right] = \mathbb{E}_{h \sim \nu_G}\left[f(hx)\right] = \bar{f}(x) \ ,$$

where $h = gu$. In practice, we will consider sets of transformations $G$ which do not necessarily form a group (e.g. rotations within a given range). Because of this, $\bar{f}$ is only approximately invariant, although it can get very close to perfect invariance, as shown in our experiments from Section 4.3.

## 3.2 Architecture: augment, forward and aggregate

As the averaged model (3.1) is intractable, one can approximate it with an empirical average

$$\tilde{f}(x) = \frac{1}{C} \sum_{c=1}^{C} f(g_c x),$$

where $g_c$'s are sampled from $\nu_G$. In practice, using a large number $C$ of sampled transformations would be prohibitive as well. Fortunately, even when $C$ is small, $\tilde{f}$ is an unbiased estimator of $\bar{f}$. Hence stochastic gradient descent can be used to train *exactly* $\bar{f}$ by minimizing the loss of $\tilde{f}$.

Based on these observations, we propose to create nearly invariant neural networks made of three blocks:

- an *augmentation module*, which takes an input $x \in \mathcal{X}$, samples $C$ transformations from $G$, and outputs $C$ transformed copies of $x$;
- a *trunk model $f$*, which can be any neural network mapping transformed inputs to predictions;
- and an *aggregation module*, which is responsible for averaging the $C$ predictions.

This general architecture is illustrated on Figure 1 and referred to as *AugNet* hereafter. Note that unlike standard data augmentation, the transformations distributions is part of the model and kept at inference.

## 3.3 Augmentation layers

There are two missing elements from the architecture described in the previous section, namely the choice of the set of transformations $G$ and how transformations are sampled from it. Indeed, the true invariances present in a dataset are often unknown, which is why we propose to learn both the set of transformations and a parametrized distribution used to sample from them.

In this paper, we are mostly interested in transformations $T : \mathcal{X} \to \mathcal{X}$ defining a data augmentation, such as random rotations in image recognition. Most often, such operations can transform the data with more or less intensity depending on a parameter $\mu$, called *magnitude* hereafter. If we take the same example of random rotations, the magnitude can be the maximum angle by which we are allowed to rotate the images. In order to have an homogeneous scale for all transformations considered, we assume without loss of generality that magnitudes lie in the interval $[0, 1]$, with $\mu = 0$ being equivalent to the identity (i.e. no augmentation) and $\mu = 1$ being the maximal transformation strength considered. By using the reparametrization trick [21] or some other type of gradient estimator such as straight-through [22] or relax [23], these transformations can be made differentiable with respect to $\mu$ as shown for example by [12] for image augmentations and by [14] for augmentations of electroencephalography signals (EEG). One can hence learn the right magnitude to use from the data by backpropagating gradients through $T$.

But the correct transformation $T$ describing a data invariance is also supposed to be unknown and needs to be learned in addition to its magnitude. Let $\mathcal{T} = \{T_1, \ldots, T_Q\}$ be a discrete set of transformations possibly describing a data invariance. We propose to use backpropagation to learn which transformation to pick from this set by using a layer consisting of a weighted average of all transformations:

$$\text{AugLayer}(x; w, \mu) = \sum_{q=1}^{Q} w_q T_q(x; \mu_q), \tag{2}$$

where the weights $w_q$ sum to 1. In practice, we optimize some other hidden weights $w'$ which pass through a softmax activation $w = \sigma(w')$. As explained in Section 3.4 and demonstrated in our experiments, this layer architecture favors a single transformation describing a correct data invariance and tune its magnitude.

As the data might be invariant to more than one transformation, these layers can be stacked on top of each other, so that each one learns a different transformation (see Figure 1). This is justified by the stability property of invariance for composition. Indeed, if a function $h : \mathcal{X} \to \mathcal{Y}$ is invariant to actions $u$ and $g$, then it must be invariant to their composition: $h((u \circ g)x) = h(u(gx)) = h(gx) = h(x)$. Hence, we build augmentation modules with sequences of augmentation layers (2) and learn their parameters $w, \mu$ from the training set together with the parameters of the trunk model.

### 3.4 Selective regularizer

As illustrated in our experiments from Section 4.1, if we train *AugNet* with a standard loss such as the cross-entropy, the model tends to find the most unconstrained model: the one with $\mu = 0$, which never augments the data. The same was described for Augerino [19], which is why its authors proposed to add a regularization $R$ pushing towards broader distributions:

$$\min_{w,\mu,\theta} \ell(\tilde{f}(x; w, \mu, \theta), y) + \lambda R(\mu) \ . \tag{3}$$

In the previous equation, $\theta$ denotes the parameters of the trunk neural network $f$ and $\ell$ is a loss function. With our formalism, the regularization proposed by [19] is equivalent to penalizing the negative $L_2$-norm of the magnitude vector: $R(\mu) = -\|\mu\|_2$. As illustrated in the ablation experiments of Section 4.1, this regularizer is not sufficient to ensure that our augmentation layers do not converge to the identity transform because we have an additional degree of freedom than Augerino: the weights $w$. Hence, the model may reach low loss values by just maximizing the weight of a single transform $w_q = 1$ with magnitude $\mu_q = 0$, while maximizing all other magnitudes $\mu_{q \neq q'} = 1$. Because of this, we propose instead to penalize the norm of the element-wise product of weights and magnitudes: $R(w, \mu) = -\|w \odot \mu\|_2$. This has the effect of tying together $w_q$ and $\mu_q$ of the same transformation $T_q$, avoiding the problem described before.

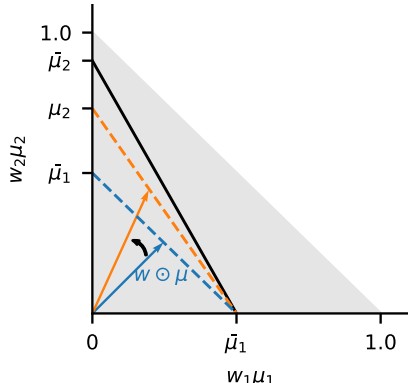

Figure 2: Illustration of the selective property of AugNet's regularizer. As the norm or the vector $w \odot \mu$ increases and $\mu_1$ reaches the maximal value allowed by the training loss $\bar{\mu}_1$, $\mu_2$ starts getting greater than $\mu_1$, creating an imbalance in the weights gradients. When $\mu_2$ reaches $\bar{\mu}_2$, the only way to keep increasing the vector's norm is by decreasing $w_1$ towards 0 and increasing $w_2$ towards 1.

Another property of this regularization is that it promotes the selection of a single transformation per augmentation layer, as illustrated on Figure 2 for the simple case of $n = 2$ transformations. Because $\sum_q w_q = w_1 + w_2 = 1$, the vector $w \odot \mu = (x_1, x_2) \in [0, 1]^2$ is bound to move within the line of equation $x_2 = \mu_2(1 - \frac{x_1}{\mu_1})$ when the magnitudes $\mu_1$ and $\mu_2$ are fixed. At the beginning of the training, with all weights initialized at $0.5$ and magnitudes close to $0$, if neither $T_1$ or $T_2$ harm the training, the vector's norm can grow following the bissector by increasing the magnitudes equally. Once one of the transformations $T_1$ reaches a magnitude $\bar{\mu}_1$, it will start harming the training loss and its gradient will converge to 0. This introduces an imbalance between the transformations as $\mu_2$ keeps increasing, until it reaches its maximal value $\bar{\mu}_2 > \bar{\mu}_1$. As $w \odot \mu$ is still bound to move within the line $\bar{\mu}_2(1 - \frac{x_1}{\bar{\mu}_1})$, the only way to keep increasing its norm is by pointing more and more vertically, until it gets to $w_2 = 1$ and $w_1 = 0$. This can also be seen from the expression of the gradients of the regularizer:

$$\frac{\partial R^2}{\partial w_i}(w, \bar{\mu}) = 2\bar{\mu}_i^2 - 2w_i(\bar{\mu}_1^2 + \bar{\mu}_2^2) \ .$$

which implies that:

$$\nabla_{w_1} R^2(w, \bar{\mu}) \le \nabla_{w_2} R^2(w, \bar{\mu}) \iff \bar{\mu}_1 \le \bar{\mu}_2 \ .$$

Each augmentation layer hence promotes the transformation with the highest admissible magnitude.

## 4 Experiments with synthetic data

In this section we present experimental results of *AugNet* in controlled settings in order to verify empirically some of its properties and compare it to *Augerino* and standard data augmentations.

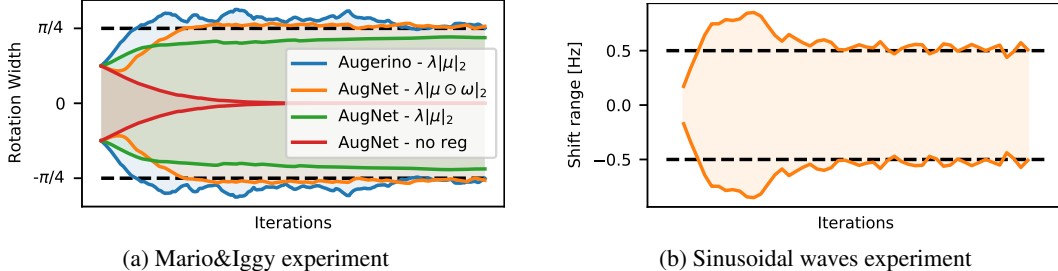

(a) Mario&Iggy experiment         (b) Sinusoidal waves experiment

Figure 3: **(a)** Learned rotation angle during training. Both Augerino [19] (blue) and AugNet (orange) are able to learn the correct level of rotation invariance in the data. When the regularization is removed from AugNet (red), it converges to the identity transform (no rotation) as observed in [19]. If we replace the regularizer in AugNet by Augerino's one, AugNet misses the correct rotation angle (green). **(b)** Learned frequency shift invariance. AugNet is able to learn nonlinear invariances from various types of data.

## 4.1 Comparison to Augerino

**Learning the correct invariance**  First, we reproduced the Mario-Iggy experiment by [19] to show that AugNet can also learn affine invariances from image datasets. The data is generated from two initial images, which are rotated by a random angle either between $[-\pi/4, \pi/4]$ (labels 0 and 2) or between $[-\pi, -3\pi/4] \bigcup [3\pi/4, \pi]$ (labels 1 and 3). This procedure is illustrated in Figure A.1. Hence, labels depend both on the initial image and on whether it has its head pointing up or down. By design, the data is invariant to inputs rotations between $[-\pi/4, \pi/4]$, and the experiments objective is to verify whether AugNet can learn this invariance. For this, we use a single augmentation layer containing 5 geometric augmentations: `translate-x`, `translate-y`, `rotate`, `shear-x` and `shear-y`. Note that while these augmentations are also encoded in Augerino, we don't use the Lie algebra and exponential maps to implement them here. Maximal translations are set to half the image width/height, maximal rotations correspond to an angle of $\pm\pi$ and maximal shearing coefficients are 0.3. The trunk model used is a simple 5-layer convolutional network, whose architecture is described in Section A.1 together with other experimental details. As shown on Figure 3a, both Augerino (blue) and AugNet (orange) are capable of learning the correct angle of invariance from the data.

**Ablation study**  As explained in Section 3.4, the regularizer plays a crucial role both in AugNet and Augerino. It can be seen on Figure 3a that without regularization the rotation angle converges to 0. Indeed, our model naturally tends to nullify any transformations of the input, similarly to what is shown by [19] for Augerino. Furthermore, we also see that when we replace AugNet's regularizer by Augerino's one ($-\|\mu\|_2$), the model does not learn the correct angle. This happens because this penalty is not sufficient to prevent the model from converging to the identity transform, as shown by the learned weights and magnitudes depicted at the bottom of Figure 4. We see indeed that the model maximizes the weight for `translate-x` and minimizes its magnitude, while maximizing the magnitude of other transformations which have very small weights. In contrast, we can see at the top of Figure 4 that the regularizer we propose allows to both *select* the `rotate` operation and correctly tune its magnitude.

## 4.2 Application to problems beyond computer vision

In this experiment we demonstrate that AugNet is also applicable to other types of data and can learn non-linear transformations which go beyond the scope of Augerino. For this, we create a dataset consisting of four classes corresponding to 4 generating frequencies: $\omega = 2, 4, 6$ and 8 Hz. For each of these frequencies, 10 seconds long sinusoidal waves of unit amplitude are created with frequencies sampled uniformly between $\omega \pm 0.5$ Hz. Moreover, the phase of these waves are sampled uniformly and they are corrupted with additive Gaussian white noise with a standard-deviation of 0.5 (*cf.* Figure A.4). We generate 400 training examples and 200 test examples. The learning task consists in predicting the correct generating frequency $\omega$ from the noisy signals. Note that, because classes are separated by 1 Hz, the dataset presents an invariance to frequency shifts of the inputs between $\pm 0.5$ Hz.

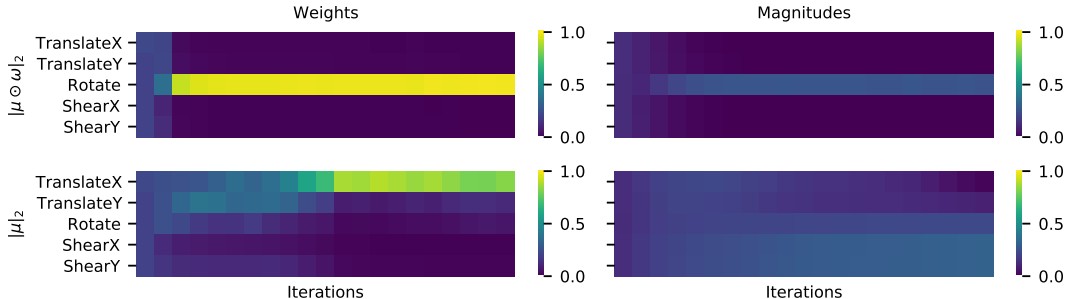

Figure 4: **Top:** With its standard regularizer, AugNet quickly learns to maximize the probability weight of rotation and adjusts the corresponding magnitude, ignoring other irrelevant transformations. **Bottom:** With Augerino's regularizer, the model may still converge to the identity map by maximizing the probability weight of any (irrelevant) transformation and minimizing its magnitude towards 0.

In order to learn this invariance, we use a single augmentation layer implementing three augmentations from the EEG literature: `frequency shift` [14], `FT surrogate` [24] and `Gaussian noise` [25]. The trunk model used is a simple 3-layer convolutional network described in Section A.2 together with other experimental settings. As shown in Figure 3b, AugNet is able to learn the correct range of invariance, and select the correct transformation. This experiment was repeated in Section B.1 with multiple augmentation layers to show that AugNet is robust to the chosen number of layers.

### 4.3 Insensitivity to model capacity

In this experiment we aim to demonstrate that models trained with our proposed methodology are invariant regardless of the capacity of the trunk model $f$ used, which represents an advantage over (automatic) data augmentation. For this, we re-use the sinusoids dataset from Section 4.2 and compare three methods: a baseline model trained directly on the raw data, the same model trained with an oracle augmentation $T$ (`frequency shift` with bounds of $\pm 0.5$ Hz) and an AugNet using the baseline as trunk model. Unlike in Section 4.2, we use here a very small multi-layer perceptron with a varying number of neurons and layers, in order to be able to assess the impact of the model's expressivity.

In addition to the generalization power of the trained models, evaluated through their test accuracy, we are interested in how invariant they are to the true symmetry $T$ encoded in the data. To gauge this property for a model $f$ at some input $x \in \mathcal{X}$, we use the following metric adapted from [26]:

$$\text{Inv}_T(f(x)) = \frac{b - d(f(x), f(T(x)))}{b} \quad , \quad (4)$$

where $d$ is the cosine distance and $b = d(f(x_i), f(x_j))$ is a baseline distance

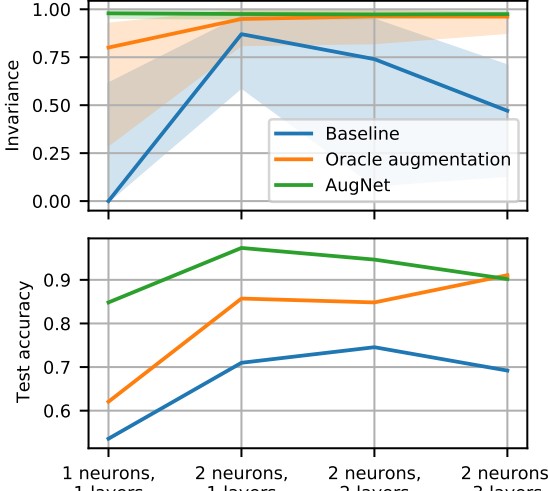

Figure 5: **Top:** Model invariance (4) to the true frequency shift in the data across different architectures. We report the median values across the test set, with a 75% confidence interval. Data augmentation helps to increase the model invariance, but is limited by its expressivity, even when data invariances are known. AugNet *learns* to be almost perfectly invariant and is insensitive to model capacity. **Bottom:** AugNet outperforms a model trained with *oracle* augmentation with 3x fewer parameters.

between randomly shuffled inputs [1]. As applying $T$ to the inputs of an invariant model should leave its outputs unchanged, the closer $\text{Inv}_T(f(x))$ is to 1 for various inputs $x$, the more $f$ is invariant to $T$.

---

[1] Here $f$ refers to the activations before the final softmax.

Figure 5 reports the median invariance across the test set of the three models for different number of layers and neurons. It can be seen that data augmentation helps to increase the level of invariance of the baseline model considerably. However, the level of invariance reached for the smallest models is lower than for larger models, suggesting that the former cannot encode perfectly the invariance taught to them through data augmentation. This problem would be worse in a realistic setting where augmentations are more complex and imperfectly learned by ADA approaches rather than known *a priori*. On the contrary, results show that AugNet is capable of learning the correct augmentation from scratch and make the baseline model almost perfectly invariant to it, regardless of its expressivity. Furthermore, the plot at the bottom of Figure 5 suggests that the level of invariance of each model is tightly related to its predictive performance.

## 5 Experiments with real datasets

### 5.1 Image recognition

In this experiment we showcase AugNet on a standard image recognition task using the CIFAR10 dataset [27]. All models considered in this experiment are trained for 300 epochs over 5 different seeds on a random 80% fraction of the official CIFAR10 training set. The remaining 20% is used as a validation set for early-stopping and choosing hyperparameters, and the official test set is used for reporting performance metrics. The trunk model used here is a pre-activated ResNet18 [28, 29].

We consider five baselines in this experiment: Augerino [19], AutoAugment [6] (pretrained for 5000h), RandAugment [30], as well as the trunk model trained with and without fixed data augmentations (`horizontal-flip` with probability 0.5 and `random-crop` of 32x32 pixels with a padding of 4 pixels [2]). For Augerino, AugNet and the baseline with fixed augmentations, we apply a standard normalization of RGB channels to the augmented examples before feeding them to the trunk model. AugNet's augmentation module is made of two layers containing the five augmentations detailed in Section 4.1 and a third layer with the following four different augmentations: `horizontal-flip`, `sample-pairing` [31], `contrast`, `brightness jitter` [32]. Both AugNet and Augerino were trained with a number of copies set to $C = 1$ and tested with $C = 20$ to take advantage of the invariant averaging architecture. Contrary to [6, 30], we do not add the learned augmentations on top of the fixed baseline transformations. The reader is referred to Section A.3 for further experimental details, and to Section B.2 for a sensitivity analysis on hyperparameters $C$ and $\lambda$.

Figure 6a reports the performance across epochs. For a given epoch, we report the test accuracy corresponding to the best validation accuracy up to that point in training (equivalent to having early-stopped the model). The performance of the baseline model with no augmentation ($86.6 \pm 0\%$) is considerably improved by Augerino ($91.6 \pm 0.1\%$), AutoAugment ($92.0 \pm 0.1\%$), RandAugment ($92.4 \pm 0.2\%$) and AugNet ($93.2 \pm 0.4\%$). AugNet consistently outperforms all three baselines over all seeds within a 90% confidence interval. This is probably due to AugNet selecting transformations which cannot be learned by Augerino, such as the `horizontal flip` (*cf.* Figure B.7), and to the better invariance of AugNet compared to models trained with data augmentation (*cf.* Figure B.9). Concerning the baseline with fixed augmentations ($93.6 \pm 0.2\%$), AugNet reaches a comparable performance. This demonstrates that in a realistic scenario with data from a real-world application where augmentations have not been explored as extensively as for CIFAR10, AugNet can be trained end-to-end and lead to a performance comparable to what we would get after manually trying many augmentation combinations.

### 5.2 Sleep stage classification

Similarly to Section 4.2, this experiment aims to demonstrate the practical usefulness of AugNet in a realistic setting beyond affine transformations. For this, a sleep stage classification task is considered. As most commonly done [33], it consists in assigning to windows of 30 s of EEG signals a label among five: Wake (W), Rapid Eye Movement (REM) and Non-REM of depth 1, 2 or 3 (N1, N2, N3). The public dataset MASS - Session 3 [34] is used for this purpose (more details in Section A.4). As done by [14], both the training and validation sets consist of 24 nights each, and the test set contained 12 nights. The trunk model used for this experiment is the convolutional network proposed in [4], whose architecture is detailed in section Section A.4 together with other experimental settings. In this experiment, we compare the proposed approach to ADA methods. For this, two discrete search methods were considered: *AutoAugment* [6] and *Fast AutoAugment* [11]. Additionally, three gradient-based methods were also tested: *Faster AutoAugment* [12], *DADA* [13] and *ADDA* [14]. All these methods shared the same policy architecture, consisting in 5 subpolicies made of 2 augmentation transformations (*cf.* Section A.4). As for the previous experiment, we used two types of augmentation

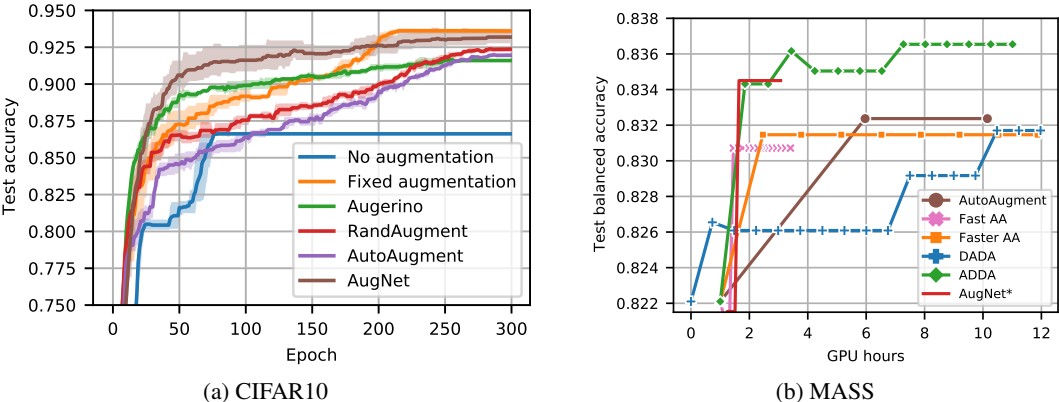

(a) CIFAR10            (b) MASS

Figure 6: **(a)** Median test accuracy on CIFAR10, over 5 seeds, where shades represent 90% confidence. AugNet learns invariances from scratch, outperforms automatic search approaches and achieves a performance comparable to state-of-the-art fixed data augmentations. **(b)** Median performance (over 5 folds) on the MASS dataset, as a function of the computation time. AugNet is comparable in speed and accuracy to ADDA, while being trained end-to-end. This figure is also available with error bars in the appendix Figure B.10

layers to implement AugNet: two layers capable of sampling *time-frequency* transformations, and one layer made of *sensors* transformations (*cf.* Table A.4).

Figure 6b presents the balanced accuracy over the test set as a function of the computation time. For ADA methods, we stopped the search every given number of steps (2 epochs or 5 samplings), used the learned policy to retrain the model from scratch (leading to a point in Figure 6b) and resumed the search from where it stopped. For each run, we report the test accuracy of the best retrained model according to the validation accuracy. Also note that some methods (Fast AutoAugment and ADDA) require a pretraining of the model. Hence, the reported GPU hours for ADA methods correspond to the sum of pretraining, search and retraining time, which explains why they start with some horizontal offset. We see that, given a budget of 12 hours, AugNet is able to outperform four out of the five state-of-the-art approaches both in speed and accuracy. It reaches a final performance comparable to the recently proposed ADDA, while requiring considerably less efforts in parameter fine tuning. AugNet only requires to set two hyperparameters: the regularization parameter $\lambda$ which is only one-dimensional, and the number augmentation layers stacked in the network. In contrast bilevel approaches require to carefully tune jointly the learning rate and the batch size of the validation set for the outer problem. Moreover, while AugNet is trained *end-to-end* once on the training set, ADDA requires pre-training the model and a final retraining of the model with the learned augmentation policy.

## Conclusion

In this paper we propose a new method coined AugNet to learn data invariances from the training data, thanks to differentiable data augmentation layers embedded in a deep neural network. Our method can incorporate any type of differentiable augmentations and is applicable to a broad class of learning problems. We show that our approach can correctly select the transformation to which the data is invariant, and learn the true range of invariance, even for nonlinear operations. While (automatic) data augmentation is limited by the capacity of the model to encode symmetries, our approach leads to almost perfect invariance regardless of the model size. On computer vision tasks, this advantage allows our method to reach high generalization power without hand-crafted data augmentations. Promising results are also obtained for sleep stage classification, where AugNet outperforms most ADA approaches both in speed and final performance with *end-to-end* training, avoiding tedious bilevel setup.

**Limitations** In order to be able reach top performance, a large number of copies $C$ might be necessary, increasing AugNet's computational complexity at test time (*cf.* Section B.2). This means that a proper trade-off between performance and speed has to be determined for each use-case. Moreover, although AugNet is able to learn augmentations in a broader scope than previous end-to-end methods, it is still limited to augmentations that can be relaxed into differentiable surrogates and requires the availability of a set of possible augmentations $\mathcal{T}$. These limitations are not specific to AugNet however, as it is a standard requirement in ADA [6, 12, 13, 11].

## Acknowledgments and Disclosure of Funding

This work was supported by the BrAIN grant (ANR-20-CHIA-0016) and ANR AI-Cog grants (ANR-20-IADJ-0002). It was also granted access to the HPC resources of IDRIS under the allocation 2021-AD011012284R1 and 2022-AD011011172R2 made by GENCI.

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
