# A  Experimental settings

In all experiments, a grid-search was used to find the value of the regularization parameter $\lambda$ from (3). In most cases, the grid of values used was $0.05, 0.1, 0.5$ and $1.$. For the sleep stage classification experiment of Section 5.2, stronger values were explored: $1, 5, 10, 50$. An analysis for the CIFAR10 experiment of Section 5.1 can be found in Section B.2.

## A.1  Mario-Iggy experiment – Section 4.1

The data was generated as described in Figure A.1 and Section 4.1. We used 10000 training examples and 5000 test examples, and sampled batches of 128 images, as in [19]. The trunk model used for both Augerino and Augnet are described in Table A.1. The official code from [19] was used for this experiment. As in the original experiments, all models were trained for 20 epochs using Adam [37], with $\beta_1 = 0.9$ and $\beta_2 = 0.999$. The learning rate for Augerino was set to $10^{-2}$ and weight decay was set to 0. The regularization parameter was set to $\lambda = 0.05$ (medium according to [19]). For AugNet, we used a learning rate of $5 \times 10^{-4}$ and a regularization parameter of $0.5$, together with weight decay of $1$. Augmentations were all initialized with magnitudes equivalent to $\pi/8$ for AugNet and Augerino. Augmentations were also initialized with uniform weights for AugNet.

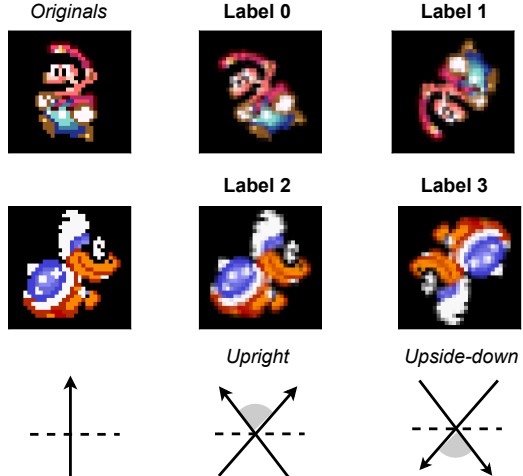

Figure A.1: Illustration of the data generation process for the Mario-Iggy experiment.

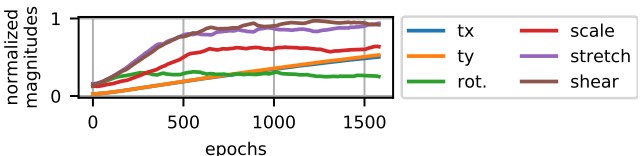

Figure A.2: Magnitudes learned by Augerino are not sparse.

## A.2  Sinusoids experiment – Section 4.2

The data was generated as described in Figure A.4 and Section 4.2. We used 400 training examples and 200 test examples, and sampled batches of 32 waves. The trunk model used for Augnet is described in Table A.2. All models were trained for 50 epochs using Adam [37]. We used a learning rate of $10^{-2}$ and a regularization parameter of $0.2$, together with weight decay of $10^{-4}$. Augmentations were all initialized with magnitudes $\mu = 0$ and uniform weights. For the experiment of Section 4.3, we set the number of copies to $C = 10$ at inference, while it was set to $C = 4$ for all other experiments. Moreover, $\lambda$ was set to $0.8$ and initial magnitudes set to $0.05$ in Section 4.3.

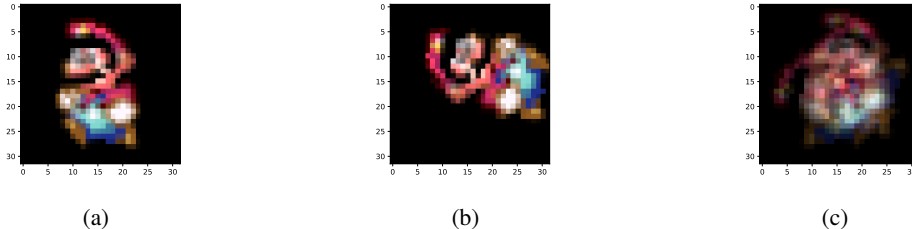

| | | | |
|---|---|---|---|
| (a) | (b) | (c) |

Figure A.3: **(a)** Original picture, **(b)** picture transformed with a sequence of `flip`, `rotate`, `crop`, **(c)** picture transformed with a convex sum of them.

| | layer | # filters | size | stride | batch norm |
|---|---|---|---|---|---|
| 1 | Conv2D | 32 | 3 | 1 | yes |
| | ReLU | | | | |
| 2 | Conv2D | 64 | 3 | 1 | yes |
| | ReLU | | | | |
| | MaxPool | | 2 | | |
| 3 | Conv2D | 128 | 3 | 1 | yes |
| | ReLU | | | | |
| | MaxPool | | 2 | | |
| 4 | Conv2D | 256 | 3 | 1 | yes |
| | ReLU | | | | |
| | MaxPool | | 2 | | |
| 5 | MaxPool | | 4 | 1 | |
| | FC | | | | |
| | Softmax | | | | |

Table A.1: Convolutional neural network architecture used in experiments of Section 4.1.

## A.3 CIFAR10 experiment – Section 5.1

The official code from [19] was used for this experiment. It is worth noting that Augerino's official code used a smaller trunk model with 13 layers, which led to poor performances. We hence replaced it by a Pre-activate ResNet18, which significantly improved performances. The official code also did not implement any cosine annealing despite what was reported in the paper [19]. We added it in our experiments but were still not able to reproduce their results exactly despite these efforts.

Following [19], we used batches of 128 images. All models were trained for 300 epochs using Adam [37] with decoupled weight-decay [38]. Also, as described in [19], we used a cosine annealing scheduler with period $T = 300$ epochs. For AugNet, Augerino and the baseline with fixed augmentations, the following normalization was used as a preprocessing on augmented data for the whole dataset: centering by $(0.485, 0.456, 0.406)$ and scaling by $(0.229, 0.224, 0.225)$. All models were trained with a learning rate set to $10^{-3}$. Weight-decay was globally set to $2 \times 10^{-2}$ for the trunk model and 0 for the augmentation modules of both Augerino and AugNet. The regularization parameter was set to $\lambda = 0.05$ for Augerino as in [19], while AugNet's regularization was set to $0.5$ for the first 40 epochs and scaled down to $0.05$ for the rest of the training. Augmentations were all initialized with magnitudes set to $0.5$ for both Augerino and AugNet, with uniform weights for the latter. In this experiments, differentiable augmentations were implemented using the KORNIA package [39], as well as the official code of Faster AutoAugment [12]. All trainings were carried on single Tesla V100 GPUs. Contrary to what is done in the official AutoAugment [6] and RandAugment [30] papers, we did not add augmentation policies on top of baseline augmentations, which explains why our performances are lower than those reported. We do this because we want to learn augmentations completely from scratch, without relying on prior knowledge of which baseline augmentations work well for a given dataset.

| | layer | # filters | size | stride | batch norm |
|---|---|---|---|---|---|
| 1 | Conv2D | 2 | 3 | 1 | yes |
| | ReLU | | | | |
| 2 | Conv2D | 2 | 3 | 1 | yes |
| | ReLU | | | | |
| | MaxPool | | 2 | | |
| 3 | GlobalPool | | | | |
| | FC | | | | |
| | Softmax | | | | |

Table A.2: Convolutional neural network architecture used in experiments of Section 4.2

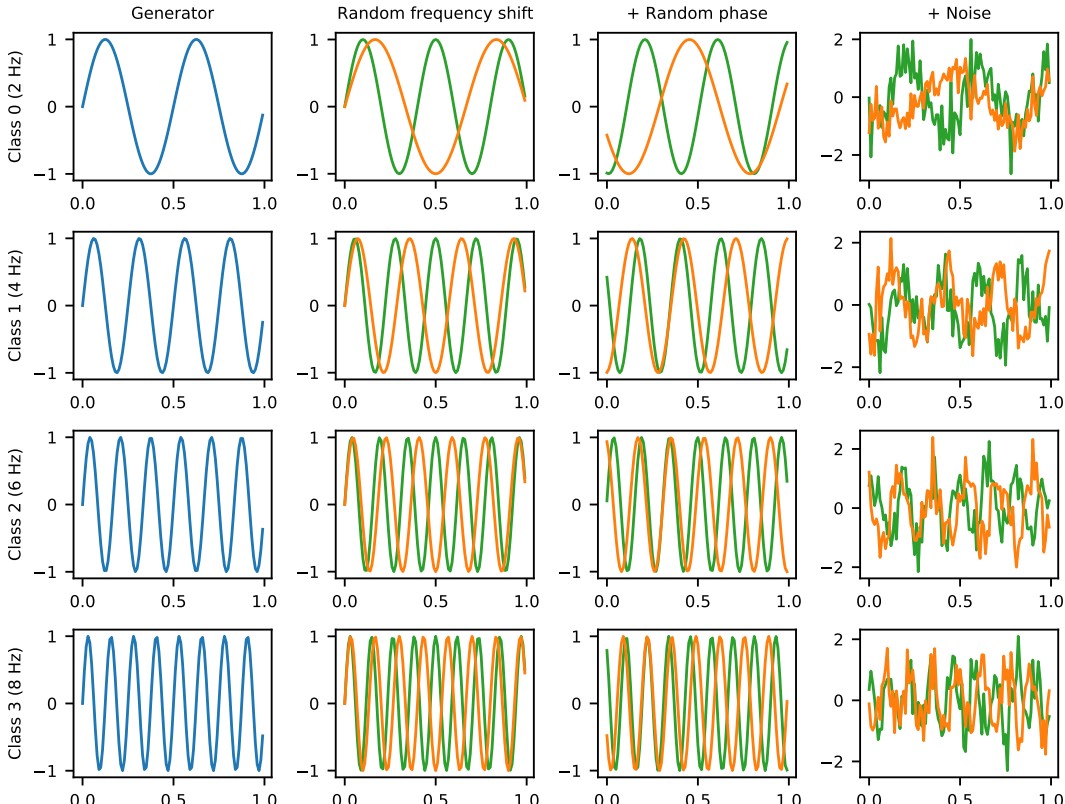

Figure A.4: Illustration (cropped to 1 second) of waves composing each class of the simulated dataset used in the sinusoids and model capacity experiments.

## A.4  MASS experiment – Section 5.2

**Dataset**  The public dataset MASS - Session 3 [34] was used for this purpose (more details in Section A.4). It corresponds to 62 nights, each one coming from a different subject. Out of the 20 available EEG channels, referenced with respect to the A2 electrode, we used 6 (C3, C4, F3, F4, O1, O2). As done by [14], both the training and validation sets consisted of 24 nights each, and the test set contained 12 nights.

**Architecture**  For all EEG experiments, learning was carried using the convolutional network proposed in [4], whose architecture is described on Table A.3. The initial number of channels $C$ was set to 8. The first layers (1-4) implements a spatial filter, computing virtual channels through a linear combination of the original input channels. Then, layers 5 to 9 correspond to a standard convolutional feature extractor and last layers implement a simple classifier.

| | layer | # filters | # params | size | stride | output dim. | activation |
|---|---|---|---|---|---|---|---|
| 1 | Input | | | | | (C, T) | |
| 2 | Reshape | | | | | (C, T, 1) | |
| 3 | Conv2D | C | C * C | (C, 1) | (1, 1) | (1, T, C) | Linear |
| 4 | Permute | | | | | (C, T, 1) | |
| 5 | Conv2D | 8 | 8 * 64 + 8 | (1, 64) | (1, 1) | (C, T, 8) | Relu |
| 6 | Maxpool2D | | | (1, 16) | (1, 16) | (C, T // 16, 8) | |
| 7 | Conv2D | 8 | 8 * 8 * 64 + 8 | (1, 64) | (1, 1) | (C, T // 16, 8) | Relu |
| 8 | Maxpool2D | | | (1, 16) | (1, 16) | (C, T // 256, 8) | |
| 9 | Flatten | | | | | (C * (T // 256) * 8) | |
| 10 | Dropout (50%) | | | | | (C * (T // 256) * 8) | |
| 11 | Dense | | 5 * (C * T // 256 * 8) | | | 5 | Softmax |

Table A.3: Detailed architecture from [4], where $C$ is the number of EEG channels and $T$ the time series length.

**Training hyperparameters** The optimizer used for all models was Adam with a learning rate of $10^{-3}$, $\beta_1 = 0.$ and $\beta_2 = 0.999$. At most 300 epochs were used for training, with a batch size of 16. Early stopping was implemented with a patience of 30 epochs. For ADDA, the policy learning rate was set to $5 \times 10^4$ based on a grid-search carried using the validation set. For AugNet, the regularization parameter was set to $\lambda = 10$. Balanced accuracy was used as performance metric using the inverse of original class frequencies as balancing weights. The MNE-PYTHON [35] and BRAINDECODE software [36] were used to preprocess and learn on the EEG data. Training was carried on single Tesla V100 GPUs.

**Augmentations considered** The 13 operations considered are listed in Table A.4. A detailed explanation of their implementation can be found in the appendix of [14]. While all this augmentations were used by gradient-free algorithms, `bandstop filter` was not included in the differentiable strategies (Faster AA, DADA, ADDA, AugNet) because we did not implement a differentiable relaxation of it. All augmentations used came from the BRAINDECODE package [36]. AutoAugment was implemented replacing the PPO searcher with a TPE searcher (as in Fast AutoAugment). The OPTUNA package was used for that matter [40].

| type | transformation | range |
|---|---|---|
| Time | time reverse | |
| | time masking | 0-200 samples |
| | Gaussian noise | 0-0.2 std |
| Frequency | FT-surrogate | $0-2\pi$ |
| | frequency shift | 0-5 Hz |
| | bandstop filter | 0-2 Hz |
| Sensors | sign flip | |
| | channels symmetry | |
| | channels shuffle | 0-1 |
| | channels dropout | 0-1 |
| | rotations x-y-z | 0-30 degrees |

Table A.4: Augmentations considered in experiment 5.2. The range column corresponds to the values to which are mapped the magnitudes $\mu = 0$ and $\mu = 1$.

# B  Complementary results

## B.1  Sinusoids experiments – Section 4.2 and 4.3

Hereafter we provide further results and extensions to the experiment of Sections 4.2 and 4.3.

One may find the evolution of the augmentation module weights $w$ during training in the top-frame of Figure B.1a. We see that the correct `FrequencyShift` invariance is selected after only one epoch, its magnitude being correctly tuned as already shown on Figure 3b.

**Sensitivity to number of layers**   Since the number of augmentation layers is a hyperparameter, we may wonder whether Augnet is robust to its value. Namely, we want to know what would happen if we trained an Augnet model with more augmentation layers than there are invariances in the data. Hence, we repeated the experiment from Section 4.2 with 2 and 4 layers and plotted results on Figures B.2 and B.3 respectively. The same training data and hyperparameters were used (*cf.* Section A.2). We see in both cases that all layers select the correct transformation and only one gets a positive magnitude $\mu_i > 0$, all other layers converging to the identity. These results suggest that Augnet is indeed robust to the number of layers chosen and can correctly learn underlying invariances even when we decide to use "too many" of them.

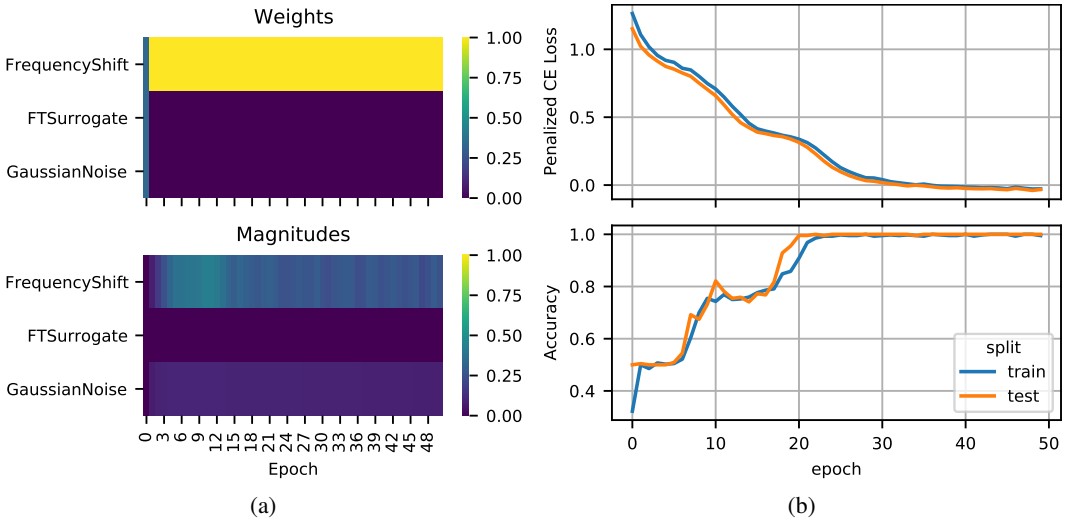

Figure B.1: **(a)** Evolution of learned weights and magnitudes in the sinusoids experiment Section 4.2. AugNet quickly learns to maximize the frequency shift invariance and drop the others. **(b)** Loss and accuracy during AugNet training in the sinusoids experiment Section 4.2.

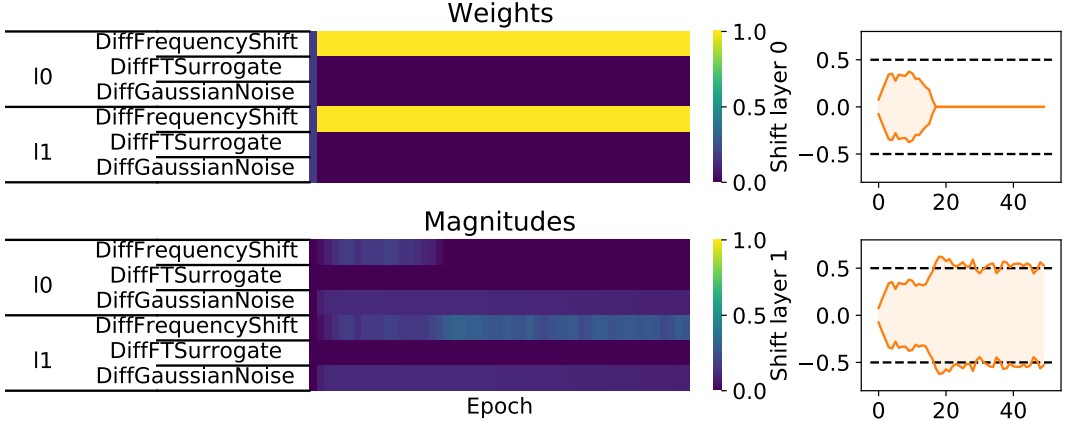

Figure B.2: Weights and magnitudes learned by a model with 2 augmentation layers.

**Sensitivity to $C$**   Another important hyperparameter to study in the context of this experiment is the number of copies $C$. Indeed, this parameter intuitively defines how well AugNet approximates the expectation in proposition 3.1, and hence how invariant it is to the learned transformations. Figure B.4 is an extended version of Figure 5, where we have tested three different values $C$ at inference: 1, 4 and 10. It confirms the intuition that the greater $C$, the more invariant is AugNet, and demonstrates that this has hence an important impact on performance. The case $C = 1$ works as a "sanity check"

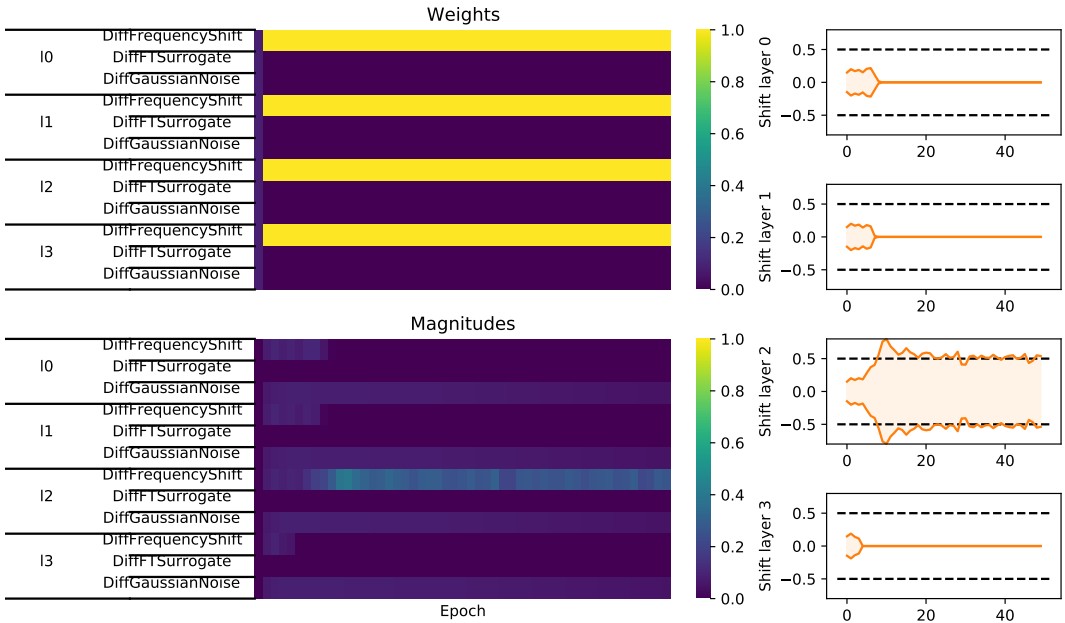

Figure B.3: Weights and magnitudes learned by a model with 4 augmentation layers.

showing that we cannot do much better than the baseline with no augmentation if we don't average the model's predictions.

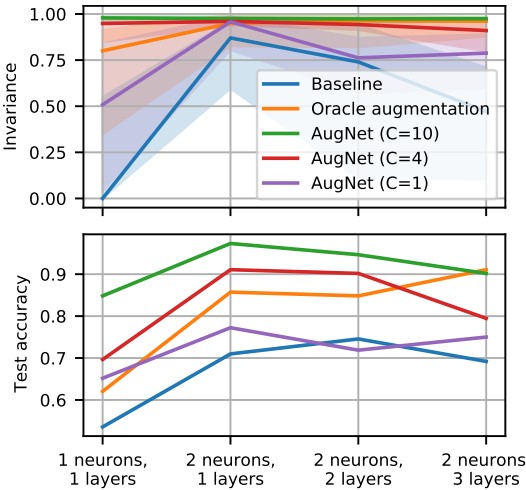

Figure B.4: **Top:** Model invariance (4) to the true frequency shift in the data across different architectures. We report the median values across the test set, with a 75% confidence interval. The greater $C$, the more invariant is AugNet. **Bottom:** By controlling the invariance, $C$ also controls AugNet's accuracy. AugNet has a performance close to the baseline with no augmentation if we don't average the model's predictions ($C = 1$).

## B.2 CIFAR10 experiment – Section 5.1

**Sensitivity analysis to $C$ and $\lambda$**    AugNet introduces two new hyperparameters: the penalty weight $\lambda$ and the number of copies $C$. Figure B.5 shows a sensitivity analysis of the final performance on CIFAR10 (Section 5.1) for both these hyperparameters. We see on Figure B.5b here again that larger values of $C$ yield better performances, as seen for the sinusoids simulated experiment in Figure B.4.

However, increasing the number of copies $C$ at inference also comes with a computation time that increases linearly, as shown on Figure B.6.

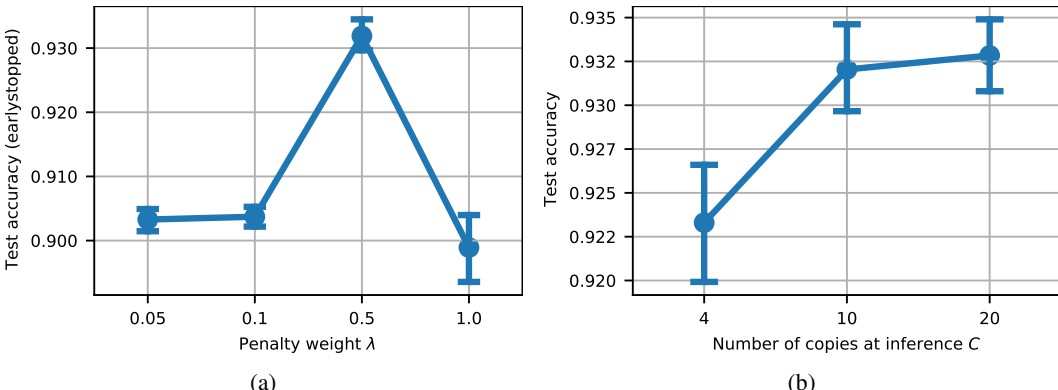

(a)                             (b)

Figure B.5: **(a)** Test accuracy on CIFAR10 for different penalty weights $\lambda$. $C$ is set to 20 in this experiment and all other details are given in Section A.3. **(b)** Test accuracy on CIFAR10 for different number of copies $C$ at inference. All points correspond to the same model trained with $C = 1$ at training and evaluated with different values of $C$ at inference.

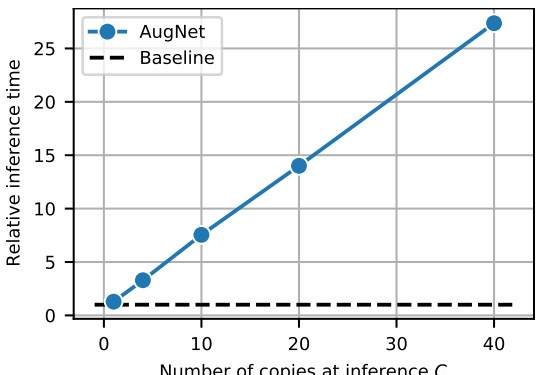

Figure B.6: Average forward-pass time for different values of $C$ at inference. Error shades of 75% confidence over batches are depicted, but very small. Results are normalized by the forward-pass time of the trunk model alone $f$, represented by the black dashed line.

**Learned augmentations and model invariance** Figures B.7 and B.8 show which invariances were learned for two of the five runs of the CIFAR10 experiment from Section 5.1. We see on the first case that layer 1 learned the `HorizontalFlip`, which is one of the two augmentations used in the experiment baseline leading to the state-of-the-art results. We indeed got the best test accuracies (93.4% and 94.0%) for the two runs which led to the selection depicted on Figure B.7 compared to 92.7 - 93.0% for the other three runs with selections similar to Figure B.8. We can also see that all runs selected the `translate-y` invariance (and three of them selected `translate-x`), which is equivalent to the `random-crop` also used by the baseline with fixed augmentation.

Regarding the fact that for some layers, the magnitude of the selected transformation drops to 0 at a certain point in training, note that this only means that the augmentation is not required anymore to ensure the necessary level of invariance. Indeed, it is known since Population-based augmentations [10] and RandAugment [30] that the best augmentation depends on the stage of training. As shown on Figure B.9, AugNet remains invariant to these augmentations after those moments, which means that the weights of the trunk model $f$ have learned the invariance and that there is hence no more need to sample it.

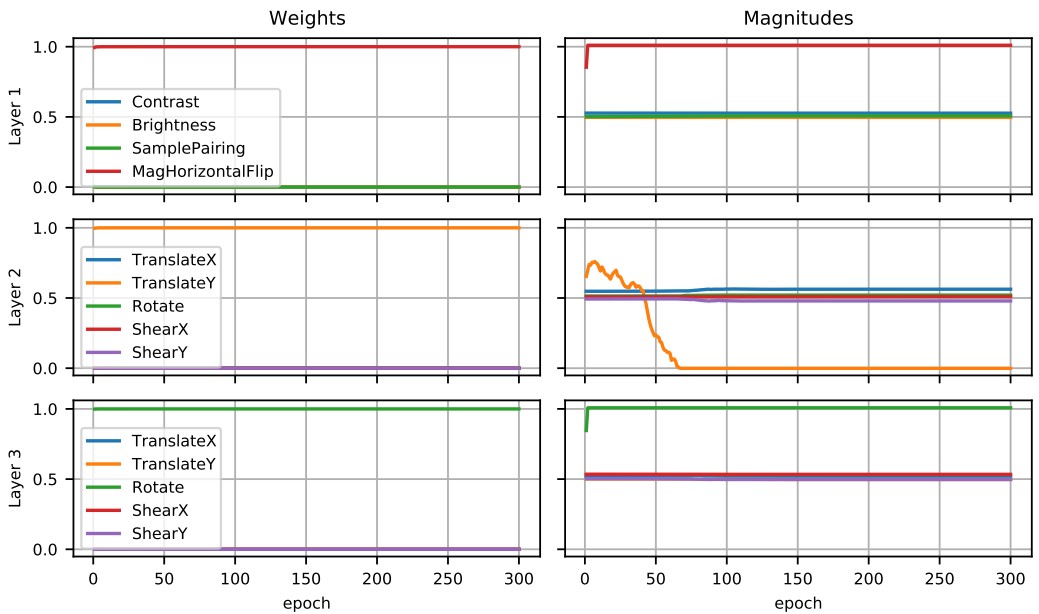

Figure B.7: Learned invariances for one run of the CIFAR10 experiment. Very similar results were obtained for another run, an both led to the best performances.

## B.3 MASS experiment – Section 5.2

As explained in Section 5.2, the plots from Figure 6b were obtained by computing median scores over 5 runs using different splits of the MASS dataset with a cross-validation scheme. In each run, the training, validation and test sets all contained data from different subjects. The same plot is presented in Figure B.10a with 75% confidence error bars. Because of the well-known large inter-subject variability inherent to EEG recordings, we see that the between-splits variance is sometimes larger than the median performance gaps between methods, making it difficult to draw strong conclusions. In an effort to circumvent the inter-subject variability issue, relative scores were computed with regard to the ADDA method after a budget of 2 hours of training and plotted in Figure B.10b. Because we are computing performance gaps independently for each split, the latter are not hidden by the between-split variance in this case. This plot shows that in the context of a small training budget, AugNet delivers better performances in sleep stage classification compared to ADDA in four out of five runs.

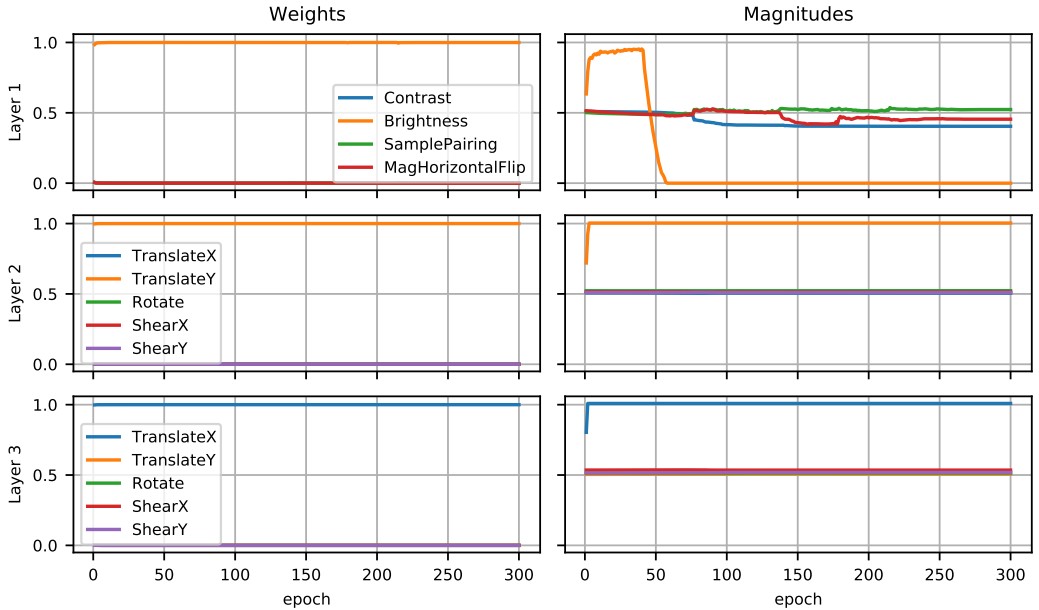

Figure B.8: Learned invariances for another run of the CIFAR10 experiment. Very similar results were obtained for three other runs.

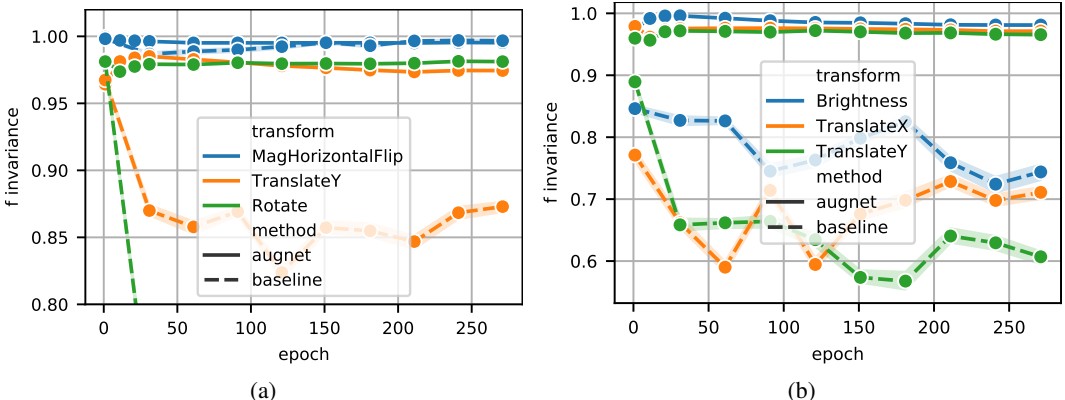

Figure B.9: **(a)** Invariance (4) of AugNet and fixed augmentation baseline to augmentations selected by AugNet during the run from Figure B.7. AugNet remains invariant to `translate-y` even after its magnitude drops to 0. **(b)** Invariance (4) of AugNet and fixed augmentation baseline to augmentations selected by AugNet during the run from Figure B.8. AugNet remains invariant to `brightness` even after its magnitude drops to 0.

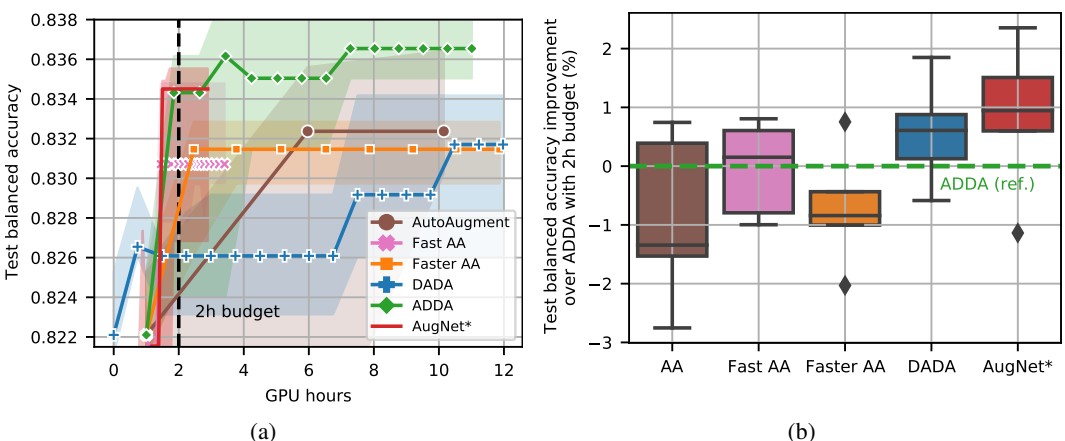

(a)  (b)

Figure B.10: **(a)** Error bars correspond to 75% confidence intervals over folds. **(b)** Fold-wise test balanced accuracy improvements with relation to ADDA after 2h of training. Performances at 2h of training were linearly interpolated from figure (a).