# OpenReview forum: "Deep invariant networks with differentiable augmentation layers"
_NeurIPS.cc/2022/Conference — NeurIPS 2022 Accept_

### Official Review · Reviewer_GmuX · 2022-07-08

**Rating:** 4
**Confidence:** 4
**Soundness:** 2 fair
**Presentation:** 3 good
**Contribution:** 2 fair

**Summary:**

The paper recognizes the difficulties of current bilevel optimization of automatic data augmentation (ADA) methods.
The authors propose AugNet, by adding learnable augmentation layers and an aggregation module to a base trunk model, and making network training and augmentation learning a single level end-to-end optimization problem.
Experiments across two simulated tasks, one image classification task, and one physiological data classification task are conducted to show the empirical versatility of AugNet.


**Questions:**

While itemized suggestions are laid out in the “weaknesses” section, here’s some higher level suggestions.

Overall I think this work is well motivated and has potential to make an impact if the experimental validation was much more convincing.

To me the first thing the authors could do without adding additional tasks and models is getting more solid results on existing tasks. As I mentioned in “weaknesses” error bars should be reported in all results to determine whether any effects are statistically significant, this should be the basis of experimental validation. It will also be nice to see stronger empirical performance across the existing tasks. As is, the improvements of AugNet over prior art is marginal and qualitative at best, with significant performance drop for Cifar. Important experiments such as performance vs lambda are missing.

The paper would also benefit a lot by going beyond the toyish task/model combinations. It is important to see whether AugNet generalizes to larger tasks and models for it to be more useful practically.


**Limitations:**

The authors discussed various limitations of previous work such as the bilevel optimization nature of ADAs. While there are mentions such as how practical transforms deviate from the group assumption, the authors did not have a clear discussion of the limitations of AugNet.

**Strengths And Weaknesses:**

Strengths:
1. Learning data augmentation policies is an important topic in training generalizable deep neural networks. The authors provide a good review of related work and identify some important limitations of previous methods such as AA, FAA, and ADDA. Thus the topic and motivation of this work are well justified.
2. Some mathematical intuitions are discussed in the paper in Section 3. Practical approximations are mentioned for Proposition 3.1.
3. Basic experimental details such as number of epochs, learning rates, and weight decay are generally well documented, choices of hyperparameters are mostly sensible (see weaknesses for exceptions).
4. Error estimates are provided for Figure 5 Top and Figure 6 (a).
5. A range of synthetic and real classification tasks are tested.

Weaknesses:
Major:
Generally, empirical evaluation is weak and unconvincing at several levels which makes it hard to believe the proposed method has much practical advantage over existing methods.

1. Although the tasks are somewhat diverse, they are all small or toy problems. It is a well known trend that data augmentation tends to work better when training data is limited and its performance boost may diminish when the training data is abundant. It is important to empirically show how well AugNet scales to larger (both in size and complexity) datasets and tasks.

2. Similar to tasks, the models used in the experiments have very low capacities (especially for Section 4.3). Not testing AugNet with larger and more realistic models would significantly limit its potential usefulness.

3. For the Mario&Iggy experiments:

  3.1. Qualitatively, both mu regularization and mu-omega regularization work well for Augerino and AugNet as they all roughly learn the optimal Pi/4 rotation.

  3.2. It is unclear how sensitive AugNet is to lambda. Experiments with different lambda are missing.

  3.3. Since C is set to 10 for Section 4.3 and 4 for other experiments, it is reasonable C is at least a somewhat important hyperparameter.
Experiments to show how C affects performance are missing.

4. The Cifar-10 experiments are very problematic in many ways.

  4.1. AugNet simply performs poorly by falling behind even the base augmentation that was developed about a decade ago (from the AlexNet paper) which has 0 GPU hours of computation cost since it was handcrafted. Spending 1 extra GPU hour to learn an inferior augmentation policy is hardly a win.

  4.2. The gap in top-1 accuracy between AugNet (91.9%) vs Fixed Augment (93.6%) is significant as it could correspond to a 2x or 3x model size difference, this shouldn’t be understated.

  4.3. Since AugNet is supposed to improve upon methods like AA, it is natural to show results with AA and RA, since Pytorch has official implementations of searched policies on Cifar and ImageNet. These methods should significantly boost performance further.

  4.4. Figure B.5 shows that layer 2 learns TraslateY but with magnitude 0, which means identity transform, which suggests the failure of the proposed regularization. The same issue can be found in Figure B.6 layer 1.

  4.5. Additionally, other learned invariances with non-zero weights have magnitudes of 1, which also suggests that AugNet is only learning binary choices for both weights and magnitudes in this task. That also contradicts with what the authors propose regarding AugNet learning both weights and (continuous) magnitude.

5. Most accuracies in the MASS task differ by less than 1%, it is hard to evaluate if any median trends observed are statistically significant without error bars.

6. The limitations of this work are not properly discussed.

Minor:
Overall I think some math can be more concise or moved to the appendix, as this paper does not focus on proposing new theories. This will free up some space to discuss the limitations of this work and potentially more experimental evaluations of AugNet.

---

> ### Author Response · Authors · 2022-08-02
> **Detailed answer**
>
> We would like to thank the reviewer for its careful and detailed review and its valuable suggestions to improve our experimental results. Also, we appreciate that the reviewer thinks we ‘’provide a **good review of related work**’’ and for  finding our ‘’work **well motivated** and [with] **potential to make an impact**’’. We hope that the detailed answers below, and that **our new and more convincing results on CIFAR10** will satisfy the reviewer.
>
> The reviewer is mainly concerned by the experimental part of the work, and we address their remarks in order:
>
> 1. The reviewer found that we only evaluate the approach with small datasets, while data augmentations ‘’work better when training data is limited and its performance boost may diminish when the training data is abundant’’. We agree with the reviewer that data augmentation is **specially useful in low-data regimes in practice**, which is **precisely the scope of our work and the reason why we don’t experiment in very large computer vision datasets** such as ImageNet. Indeed, while nearly all automatic data augmentation papers only demonstrate their approaches on large image datasets, **many important applications of DL other than computer vision are limited today because of the scarcity of labeled data**. We are convinced that it is precisely in this regime that automatic data augmentation techniques have the most potential impact. This is the case for example in medicine or neuroscience, in which datasets are relatively small and data augmentations are considerably less intuitive and not as well-studied as for image-based applications. We hence believe that it is important to demonstrate automatic data augmentation in new scopes. Also note that this work **involved reimplementing the existing methods, whose official codes are also tailored to image data. We believe these comparisons and new implementations (to be open-sourced under publication) will be beneficial for the community.**
>
> 2. Concerning the low capacity of the models considered in the sinusoids experiment of section 4.3, the latter was precisely designed to illustrate that the performance of data augmentation is limited by the capacity of the architecture being trained, unlike AugNet. For this, **we had to decrease model capacity until it could not hard-code the ground-truth invariance anymore**, which was quite low since we picked a simple interpretable invariance. This experiment had to be cast with synthetic data because we needed to know the ground-truth invariance to compute the appropriate metric and derive the oracle augmentation.  **Furthermore, we have added to section B.2 a new similar analysis with a larger dataset and model (CIFAR10) showing exactly the same phenomenon as in toy experiment 4.3.**
>
> 3.1 Concerning experiment 4.1, we are not sure to understand the reviewer’s comment: ‘’both mu regularization and mu-omega regularization work well’’. Please note that **the $\mu$ regularization proposed in Augerino does not work for AugNet’s new architecture**, which learns the identity mapping and is hence not invariant to rotations between $\pm \pi/4$. Indeed, as explained in lines 228-233, AugNet’s weights $w$ select the translate-X transform and sets its magnitude to 0 (fig.4 bottom) when using the old regularization.
>
> 3.2 We have added a sensitivity analysis of the performance on the parameter $\lambda$ in section B.2.
>
> 3.3 We have added a sensitivity analysis to the parameter $C$ in section B.2 for both the sinusoids and CIFAR10 datasets. We confirm the intuition that increasing $C$ increases the invariance of the trained model to the set of learned transformations and that this has an impact on the model performance. We have also added an empirical analysis of the time complexity vs $C$, showing that increasing $C$ also has additional cost in inference time which increases linearly with $C$. A discussion on the trade-off between performance and time complexity of our approach has been added to our new limitations section.

---

> > ### Author Response · Authors · 2022-08-02
> > **Detailes answer (2/2)**
> >
> > 4.1 **We don’t agree with the reviewer statement that the baseline augmentation corresponds to a cost of 0 GPU hours.** It is important to remember that CIFAR10 is a well-studied benchmark dataset, for which we know good performing augmentations, and that this does not correspond to real life use-cases of deep learning where one needs to learn on a fresh dataset. **Consider the case where we are not working with CIFAR10 but rather with data from a real-world application for which augmentations have not been explored as extensively** as for computer vision benchmarking datasets (like sleep stage classification for example). In these cases, we usually have augmentation candidates (because it is a known modality), but we have no idea of which one will work best given the data, model and task being solved. Our results on sections 5.1 and 5.2 show that **our method allows to train a model end-to-end and obtain a performance comparable to what we would get after trying many augmentation combinations manually**. It also shows now that it can **outperform AutoAugment trained for thousands of hours**.
> >
> > 4.2 Also related to the last point and answering your second remark, note that we have improved the performance of AugNet by increasing $C$ from 4 to 20. We have done the same to the Augerino baseline to ensure a fair comparison. This brought us to a performance comparable to the baseline with fixed augmentations (AugNet: 93.2 +- 0.4, Fixed augmentation: 93.6 +- 0.2).
> >
> > 4.3 As advised by the reviewer, **we have added RandAugment [3] and AutoAugment [4] baselines to the CIFAR10 experiment, which are outperformed by AugNet** (RandAugment: 92.4 +- 0.2, AutoAugment: 92.0 +- 0.1). We used the pytorch implementations as advised by the reviewer. Note that AutoAugment was trained over 5000 GPU hours according to [4] + 3h of training, while our method was trained end-to-end in 4h.
> >
> > 4.4 **We don’t agree that selecting magnitude zero at one of the layers after 60 epochs in figure B.5 and B.6 constitutes a failure**. First of all, please note that the model and the augmentations are trained together and that it is known since PBA [6] and RandAugment [3] that **the best augmentation to use depends on the training stage**.  Our results just show that after 60 epochs, the translate-y and brightness augmentations start to be less relevant (the convolutional network probably already learned the invariance within its weights, not needing the augmentation anymore). To justify our claim, **we have added two new figures (B9 a and b)** where we have evaluated the invariance of AugNet to the learned transformations. The plots correspond to the same trainings depicted on figures B.5 and B.6 and **show that AugNet remains invariant to the transformations whose magnitude dropped to 0.**
> >
> > 4.5 Concerning the saturated magnitudes of figures B.5 and B.6, **note that experiments from sections 4.1 and 4.2 demonstrate that AugNet does not always learn binary transformations**. But of course, nothing prevents it from learning a magnitude of 1. or 0. In fact, the value of the learned magnitude depends on how the augmentation is parametrized, which is not particular to AugNet but rather a shared feature of nearly all automatic data augmentation methods [1-7].
> >
> > 5. We have added to the supplementary a new version of **figure 6b with error bars** corresponding to 75% confidence intervals. We kept the original in the main paper since it is more readable, but added a note referencing the new version of it.
> >
> > 6. As advised, we have also **added a section discussing the limitations of our work** in further details, including:
> > - the limitation to differentiable or relaxable augmentations
> > - the need for a set of candidate augmentations
> > - the trade-off between inference time and performance
> >
> > [1] Hataya et al. (2020) Faster AutoAugment
> >
> > [2] Li et al. (2020) DADA
> >
> > [3] Cubuk et al. (2020)  RandAugment
> >
> > [4] Cubuk et al. (2019)  AutoAugment
> >
> > [5] Lim et al (2019) Fast AutoAugment
> >
> > [6] Ho et al. (2019) Population Based Augmentation
> >
> > [7] Rommel et al. (2022) CADDA

---

> > > ### Comment · Reviewer_GmuX · 2022-08-08
> > > **Response to authors**
> > >
> > > Thank you for conducting experiments, updating the paper, and addressing my concerns. I believe that the paper has improved overall hence I will increase my rating.
> > >
> > > Please see some additional comments below.
> > >
> > > Unless I’m reading it wrong, Figure 3a does show that AugNet with mu regularization (in green) indeed learns an angle very close to the true angle (with larger errors than mu-omega regularization). Later Figure 5 shows that mu regularization fails because it learned the wrong weight instead of the wrong angle. It is not fair to say in Figure 3’s caption that mu regularization fails as one can not draw that conclusion from Figure 3.
> > >
> > > While I agree that data augmentation tend to work better when training data is scarce, larger datasets/tasks still benefit greatly from data augmentation as shown by papers such as Copy-Paste augmentation, AutoAugment and RandAugment. Limiting validation to small datasets is a valid choice, but doing so also significantly limits its practical usefulness across dataset scales.
> > >
> > > While new Cifar experiments are added and performance are improved, the quantitative empirical performance on the non-toyish datasets such as Cifar and MASS (Figure 6) is not very strong as it roughly matches "fixed augmentation" in the former and ADDA in the latter. As many of the error bars in Figure B.10 (aka Figure 6b with error bars) overlap, and the effect size is small, it seems that many methods perform similarly as many of the differences may not be statistically significant.

---

> > > > ### Author Response · Authors · 2022-08-09
> > > > **Discussion**
> > > >
> > > > We thank the reviewer for their new comments and are glad that our new experiments and revised manuscript helped to convince of the relevance of our contribution. Please note however that your **rating has not been improved yet**.
> > > >
> > > > Concerning the caption of Figure 3, we now better understand what was the problem. Indeed, we agree that this is not the correct place to comment on the failing of $\mu$ regularization, and **will postpone it to Figure 5 in the revised manuscript**.
> > > >
> > > > Concerning the results on CIFAR10, note that **fixed augmentations (and AugNet) beat AutoAugment and RandAugment** official implementations. This shows that it is not easy to beat the manually crafted augmentations, which is probably why **AutoAugment and RandAugment papers only add their policies on top of the fixed augmentations**, which we don't do here. We hence still believe that showing we can **learn augmentations from scratch without prior knowledge and achieve comparable performance to fixed augmentations is a valuable contribution**.
> > > >
> > > > Concerning the large error bars in Figure B.10 for the MASS experiment, these are due to the realistic setting in which we placed ourselves. Indeed, we repeated the experiment using a cross-validation scheme where the split separating testing and training subjects changes for each run. This is important because **(i) of the high inter-subject variability characterizing this type of data modality** and **(ii) because generalizing to new subjects is essential in medical applications**. This high variability is hence transferred to the performances of all models in this experiment. On the contrary, the CIFAR10 experiment leads to small variances **for these same methods**, as it uses a common test set shared across all runs for evaluation. While we could follow a similar scheme to reduce the variance in the sleep stage classification experiment, it would not be useful for real applications and could even lead to misleading conclusions. Nonetheless, we agree with the reviewer that it is difficult to interpret the statistical significance of our results in a setting where inter-fold variance is larger than the methods’ deltas. We hence **revised figure B.10, by computing per-fold improvements of each method compared to ADDA performance**. This allows us to compare the methods fold-wise, **leading to more statistically significant results**. For a training time budget of 2h, we obtain indeed that **AugNet beats ADDA on 4 out of 5 folds, with a median accuracy improvement of 0.94%, a first quartile q1=0.60% > 0 and last quartile q3=1.51%.** The corresponding figure will be added to the supplementary material.

---

### Official Review · Reviewer_kEKu · 2022-07-11

**Rating:** 5
**Confidence:** 4
**Soundness:** 2 fair
**Presentation:** 2 fair
**Contribution:** 2 fair

**Summary:**

The authors propose a module that learns the invariances (and their magnitudes) present in a dataset while training a neural network. The module applies a convex combination of augmentations before passing to a trunk network, where the outputs are aggregated. To prevent trivial augmentations (such as identity), the authors apply regularization to the magnitude predictions of their module. The authors test their module on synthetic and real-world data, comparing to other learned invariance approaches.

**Questions:**

* Why do the authors not compare to Augerino for the sinusoidal dataset?
* Is there any way for the authors to apply their method on a dataset where a set of plausible transformations T_Q are not known beforehand? This would be closer to the problem setup of a priori learning of augmentations.
* How does AugNet compare to related methods on a larger scale image dataset, like CIFAR-10 or ImageNet?
* Why does AugNet learn to apply a single augmentation at each layer? Wouldn’t it be preferable to capture all invariances using fewer parameters?

**Limitations:**

* The problem set-up states that the authors are interested in “data invariances [that] are not always known a priori”, but the authors still rely on pre-specifying which augmentations they will apply (denoted as a set T_Q). The propose methodology seems to be limited in scenarios where a set of transformations is not known a priori.
* Without more experimental comparison and validation, it is difficult to assess the efficacy of the proposed method

**Strengths And Weaknesses:**

Strengths
* The paper is well written and clearly presents its methodology and experiments.
* The synthetic experiment shows the ability to uncover ground truth invariances from a set, and the Mario&Iggy experiment demonstrates the benefit of the selected magnitude prediction regularization.

Weaknesses
* The proposed approach is an incremental improvement to that proposed by (Augerino, Benton et al. 2020). Both rely on aggregating the output of a network across augmentation samples to approximate an expectation, and both require some form of regularization to prevent learning identity for the transformation magnitude.
* The paper is limited in experimental validation. Some experiments do not adequately compare to related methods (e.g., Augerino on the EEG dataset)
* The authors should reference (“Learning Invariances using the Marginal Likelihood,” van der Wilk et al. 2018). It seems the aggregation module proposed by the author is similar to the Bayesian approach in the mentioned work.
* Since the authors are comparing to learned augmentation approaches, they are missing many related works that should be cited from that literature. Just a few, for example: (RandAugment, Cubuk et al. 2019), (Faster AutoAugment, Hataya et al. 2019), (Cutout, DeVries et al. 2017), (Adversarial AutoAugment, Zhang et al. 2019), (AugMix, Hendrycks et al. 2020). The authors should at least include a comparison to RandAugment in their experiments, such as Figure 6b.
* Regarding test accuracy, many recent methods that have been proposed to learn an augmentation policy perform comparable or worse than the simple and computationally quick approach offered by RandAugment. Justification of learning the dataset invariances would be strengthened by also comparing to a network that implicitly learns invariances by training on RandAugment.

---

> ### Author Response · Authors · 2022-08-02
> **Detailed answer**
>
> We would like to thank the reviewer for their remark. We are glad that they found our manuscript ‘’well written’’ and appreciated our synthetic examples for showing ‘’the ability [of our method] to uncover ground truth invariances’’. We would like to thank the reviewer for suggesting more ambitious comparisons with the state-of-the-art. We hope that the answers below and the additional baselines added to CIFAR10 will convince the reviewer of the relevance of this contribution.
>
> The main concerns of the reviewer seem to relate to
>
> A) the novelty of our method compared to Augerino [1],
>
> B) the reason why we don’t compare to it on both the sinusoids experiment with synthetic data and in the experiment of section 5.2 with EEG data,
>
> C) comparisons on computer vision datasets to other automatic data augmentation techniques and larger datasets such as ‘’CIFAR10 and ImageNet’’.
>
> Less critically,
>
> D) they suggest a few references to be cited.
>
> The reviewer also asks two other questions, which are:
>
> E)  why selecting a single augmentation per layer is a good property,
>
> F) and whether it would be possible not to rely on a pre-specified set of transformations.
>
> ## Points A)
>
> Concerning the novelty of AugNet compared to Augerino, while both share the same architectural idea of sampling augmentations and averaging the model outputs, please note that:
> **Augerino can only sample affine transformations**, because its augmentation module relies on the assumption that the group of augmentations have a Lie structure (when looking in details at the code of the augmentation samplers of AugNet and Augerino one can see that they are quite different. See code in supplementary material);
> Because of the last point, **Augerino’s scope is limited to applications such as computer vision**, where these augmentations make sense (rotations, translations, shearing and scaling);
> **AugNet’s augmentation module** has a very different modular architecture based on layers, which **allows it to learn sequences of augmentations (like employed in [2;6])** in an optimal order, which was not possible with Augerino;
> Learning sequences like in [3;6] is possible because each layer of AugNet is capable of **selecting augmentations** thanks the new regularizer proposed. Again Augerino samples from a mixture while we obtain sequences of selected augmentations.
> Lastly, we have shown in our CIFAR10 experiment of section 5.1 that **AugNet does outperform Augerino**.
>
> ## Point B)
>
> Points A and B are strongly related, the former being probably the reason why the reviewer did not understand why Augerino is not compared to AugNet on all experiments. **The reason why we don’t compare to Augerino on the sinusoids experiments of sections 4.2, 4.3 and the EEG experiment of section 5.2 is that these data types are out-of-scope for Augerino. This is explained in the paper in many places**, such as in:
> - l.92-95, section 2
> - l.235-236, section 4.2
> - l.319-320, section 5.2
> - l.45-46, section 1
>
> Our work hence **copes with a major limitation of Augerino and extends their ideas to a vastly broader range of applications**, which is one of our main contributions. This is a significant contribution since the ability to learn data augmentations to deal with small datasets is a game-changer in fields such as medical applications of AI where affine augmentations are too limiting.

---

> > ### Author Response · Authors · 2022-08-02
> > **Detailed answer (2/2)**
> >
> > ## Point C)
> >
> > Concerning point C, please note that we have made a comparison on CIFAR10 (section 5.1) and that we have **added the baseline RandAugment [2]** as requested. We have also **added AutoAugment [3]** to this same experiment, which was already benchmarked on the EEG experiment of section 5.2, just as Faster AutoAugment [4]. Both new baselines are shown to be outperformed by AugNet on CIFAR10.
> > Hence, after following the suggestions of all 4 reviewers, our manuscript presents **10 different experiments** (2 benchmarks, 3 qualitative analysis of learned invariance, 1 ablation study, 3 hyperparameter sensitivity analysis, 1 invariance vs model capacity analysis) in a total of **4 datasets** (2 synthetic and 2 real) with **2 different data types** (images and signals), **23 different augmentation operations**, and a total of **7 strong SOTA baselines**, which is why other reviewers (7J9r) found that we have a ‘’great amount’’ of experiments which ‘’demonstrate the effectiveness of the proposed methods’’.
> > Now, concerning new experiments with larger computer vision datasets, we would like to stress that our main contribution and motivation are to democratize end-to-end automatic data augmentation beyond this application of ML and in particular for settings where data augmentation is the most useful, which is when the sample size is not very large. Indeed, while nearly all automatic data augmentation papers only demonstrate their approaches on computer vision examples [1-7], **there are many other fields of application where automatic data augmentation is crucial**. In medical applications of AI, such as the analysis of EEG recordings for example, labeled data is very scarce with datasets orders of magnitude smaller than Imagenet. It is also application fields where existing data augmentations are considerably less intuitive and well-studied than those for images. **Developing new automatic data augmentation that work with more varied types of data and that can boost performance on smaller datasets is the ambition of AugNet. We hence don’t believe that adding new computer vision experiments with larger datasets would help to demonstrate the practical usefulness of our contribution.** We will add a sentence to the discussion to insist on this point.
> >
> > ## Point D)
> >
> > First of all, please note that 2 out of 6 references suggested by the reviewer **were already cited in our paper**:
> > [5] is cited in line 85 of our related work section,
> > Faster AutoAugment [4] is cited in line 69 of our related work section and benchmarked in section 5.2.
> > Concerning Cutout, it is not an automatic data augmentation approach (nothing is learned) but is rather just a data augmentation operation, which is why it is not cited. Other references suggested **[6-7] will be added to section 2**.
> >
> > ## Point E)
> >
> > The reason why learning one augmentation per layer is desirable is that **augmentations are often applied sequentially** [2, 3, 4, 6]. Furthermore, this design allows to **learn the correct order of transformations**, which don’t always commute (shearing > rotating is not equivalent to rotating > shearing). Also note that if our layers converged to a non-sparse weighted sum of transformations rather than a unique transformation, they would be hard to interpret. Indeed, **while augmentations usually encode known factors of variation of the data, their mixture does not**. We invite the reviewer see **picture A.3 added to the supplementary comparing an image transformed with a sum of augmentations to another augmented with a sequence of augmentations. While the sum of augmentations lead to a ‘’blob’’ like image, sequences lead to realistic examples.**
> >
> > ## Point F)
> >
> > We agree with the reviewer that our method is limited to scenarios where some set of candidate augmentations $\mathcal{T}$ is known. But this is **not a limitation specific to our work, but rather shared by the whole automatic data augmentation field** which is based on this same assumption [1-4; 6-10]. A common real-world scenario in which these assumptions are valid are when we are dealing with a known data modality (images, audio, EEG signals, …) for which data augmentations exist, but we don’t know which one will perform best given our dataset, model and task at hand. Indeed, it is known (cf.AutoAugment [3]) that the best augmentation depends on all these aspects. This is hence **not a strong assumption for practical use**.
> >
> >
> > [1] Benton et al. (2020) Learning Invariances in Neural Networks
> >
> > [2] Cubuk et al. (2020)  RandAugment
> >
> > [3] Cubuk et al. (2019)  AutoAugment
> >
> > [4] Hataya et al. (2020) Faster AutoAugment
> >
> > [5] van der Wilk et al. (2018), Learning Invariances using the Marginal Likelihood
> >
> > [6] Zhang et al. (2019), Adversarial AutoAugment
> >
> > [7] Hendrycks et al. (2020), AugMix
> >
> > [8] Lim et al (2019) Fast AutoAugment
> >
> > [9] Li et al. (2020) DADA
> >
> > [10] Rommel et al. (2022) CADDA

---

> > > ### Comment · Reviewer_kEKu · 2022-08-07
> > > **Response to Rebuttal**
> > >
> > > Thank you to the authors for their thoughtful response to feedback.
> > >
> > > After answering some questions and adding additional baselines, I believe the paper has improved and have updated my score. There are still, however, outstanding concerns raised by other reviewers. To further improve the paper, the authors may consider (1) scaling AugNet to larger datasets or performing further analysis (e.g., test accuracy with # training sample compared to other augmentation methods) if it mainly provides benefit in small datasets and (2) suggesting techniques to learn invariances without having to specify them as differentiable augmentations beforehand (especially useful outside of CV datasets).

---

> > > > ### Author Response · Authors · 2022-08-08
> > > > **Discussion**
> > > >
> > > > We thank the reviewer for reading our response and engaging in a discussion.
> > > >
> > > > **Concerning point (2),** please note that the **whole automatic data augmentation literature until Faster AutoAugment** [4] has been concerned only by discrete search approaches which do not require differentiable augmentations [2, 3, 6, 8]. These approaches **cannot make use of gradient-based learning (and can hence not be learned end-to-end),** making them less efficient than more modern approaches [4, 9, 10]. Furthermore, as explained to other reviewers who raised this concern, **differentiable augmentations is not a strong requirement**. Indeed, standard techniques can be used to make basically any augmentation operation differentiable (see e.g. answer to reviewer 1), and the corresponding additional **overhead is non-existent** for researchers and practitioners as long as **open-source implementations** are made available. Computer vision implementations have already been made available by Faster AutoAugment authors (as well as Kornia library), and EEG implementations will be made available upon the publication of our article. **This discussion has been added to the conclusion of our revised manuscript.**
> > > >
> > > > **Concerning point (1),** making an experiment showing the number of training examples vs performance boost with our and other augmentation approaches would only demonstrate that DA helps more in lower data regimes. This is a **well-known result** demonstrated for example in [10] and is inherent to all types of data augmentation, as pointed out by reviewer 4 (GmuX). Hence **we can’t see how this additional well-known experiment could help us better understand the benefits of our method.**

---

### Official Review · Reviewer_7J9r · 2022-07-11

**Rating:** 4
**Confidence:** 4
**Soundness:** 3 good
**Presentation:** 2 fair
**Contribution:** 1 poor

**Summary:**

The author proposed an augmentation layer to learn data invariance from the training data. The proposed method is applicable to any differentiable transformation, and the augmentation layer can learn a weighted sum of different transformations that can best captures the data invariance. Sufficient experiments are conducted to verify the proposed arguments.

**Questions:**

see Strengths And Weaknesses

**Limitations:**

There is no negative social impact of this work.

**Strengths And Weaknesses:**

Pros:

The paper proposed an idea that the data invariance can be learnt in an end to  end fashion. Greatly eased the learning process comparing to bi-level optimization methods.
Great amount of experiments are conducted on both synthetic data and benchmark datasets. Experiments are well designed to demonstrate the effectiveness of proposed methods.

Cons:
Overall, I think the paper shares very similar idea comparing to the Augerino paper.  The main difference is that the proposed method applied learnable weights to multiple transformations. Thus I consider the novelty of this paper as an incremental variation of existing works.

The author proposed a selective regularizer that is different from the one used in the Augerino work. But the difference of the proposed regularizer is basically injecting the newly proposed learnable weight parameters to the regularizer proposed in Augerino, making the proposed method more like a mirror design of the existing work.

The author explored the proposed augmentation method beyond computer vision tasks and emphasized the novelty of this extension. I do not see this point as something unique for this proposed method. In fact, there shouldn’t be obstacles for other existing methods to be applied to tasks beyond computer vision. It would be great if the author can further explain on this point if I missed/misunderstand something.

some minor points:
in line 123, the author mentioned : Fortunately, even when C is small, f˜ is an unbiased estimator of f. Can the author further explain on the reason?
Figure 4 could be a bit hard to read, the position of each label could be a bit confusing. Might be better to reformat the figure.

---

> ### Author Response · Authors · 2022-08-02
> **Detailed answer**
>
> We would like to thank the reviewer for their remarks. We are glad they found our experiments to be in ‘’great amount’’ and ‘’well designed to demonstrate the effectiveness of the proposed methods’’.
>
> The main concern raised by the reviewer relates to its novelty compared to the Augerino model proposed in [1]. Before going into detailed answers we would like to stress that Augerino was indeed inspirational for our work, and we recognize its clear scientific merit. Yet, we would like to argue in what follows that with AugNet we offer a clear novel contribution.
>
> 1. First of all, we agree that Augerino and AugNet share the same architectural invariance-promoting idea of sampling $C$ augmentations, transformation copies of the input with them and averaging the model outputs. However, **Augerino’s augmentation module is very different from AugNet’s: it can only sample some affine transformations** and uses generators in the Lie algebra associated to this group. Not only is this structure restrictive (requires a Lie group architecture and to know generators), its scope is mainly restricted to “spatial” data like images or molecules positions, for which the affine augmentations considered make sense. **AugNet’s augmentation module is more general purpose since it does not rely on Lie group structures, allowing it to sample and select non-affine transformations and hence be applied to a broader class of problems**, as demonstrated in experiments 4.2, 4.3 and 5.2.
>
> 2. Furthermore, while **Augerino’s augmentation module can only sample from the joint distribution** of affine transformations considered (e.g. translate-x, translate-y, 2D-rotation, shearing and scaling according to sections 3.2 and B of [1]), our augmentation module has a modular architecture made of stacked layers. As each layer only selects one augmentation (c.f. section 4.4), **AugNet allows to learn sequences of augmentation which was not proposed by Augerino**. This is not only more in line with how data augmentation is done in practice [2-5], it also allows AugNet to learn the best order of augmentations, as demonstrated in experiments of section 5. This cannot be done with Augerino.
>
> 3. Concerning the regularization, we also cannot agree that our design consists in simply ‘’injecting the [...] learnable weight parameters to the regularizer proposed in Augerino’’. Given that we have a new learnable set of parameters in AugNet, it was not obvious how to add it to the regularization. We tried other combinations like the norm of the sum $\|\mu + w\|$ and sum of norms \|\mu\| + \|w\|, but those do not work as well as the element-wise product proposed in our manuscript. Moreover, we provide a mathematical analysis of the two beneficial properties of our regularizer in sections 4.4, and an ablation study in section 4.1. This explains and empirically demonstrates the selective property of our regularization, which Augerino’s does not have. As a case in point, we have added to the supplementary a figure presenting the magnitudes learned by Augerino on the Mario&Iggy experiment of section 4.1 (figure A.2).  We see that **Augerino’s magnitudes are not sparse and hence that it is learning a ‘’mixture’’ of augmentations difficult to interpret, while AugNet selects a single transformation per layer thanks to its regularizer** (fig. 4 top).
> 4. Concerning the use of automatic approaches like [2-7] to learn augmentations beyond computer vision, they can in principle be applied to other data modalities, as shown in experiment of section 5.2 of the manuscript. Our main contribution regarding these approaches is to be **more efficient than discrete search approaches** [2] and **trainable end-to-end**, without the need of solving bilevel optimization problems as in [5-7]. Moreover, note that while methods [2-6] can be applied to other data modalities, **the fact is that none of these papers shows any experiments beyond computer vision**. Demonstrating these techniques in more varied contexts is hence another contribution of our work and we hope it will inspire future works with broader scopes.
>
> 5. The reviewer also asked us to explain the sentence ‘’Fortunately, even when $C$ is small, $\tilde{f}$ is an unbiased estimator of $\bar{f}$’’.
> The reason is that the empirical mean $\frac{1}{N}\sum_{i=1}^N Z_i$ built using i.i.d. observations $Z_i \sim Z$ is an unbiased estimator of $\mathbb{E}(Z)$ regardless of $N$. In our case, $N=C$, $Z_i = f(g_i x)$ with $g_i \sim \nu_G$ and $\mathbb{E}(Z)=\mathbb{E}_{g \sim \nu_G}(f(gx))=\bar{f}$, where f and x are fixed for a given iteration.
> Now increasing the value of C helps to reduce the variance of the prediction, as shown now in figure 5.1 where using $C=20$ instead of $C=4$ leads to much improved results on CIFAR10. **We have also added a new analysis of the impact of $C$ on performance (and time) to section B.2**

---

> > ### Author Response · Authors · 2022-08-02
> > **References**
> >
> > [1] Benton et al. (2020) Learning Invariances in Neural Networks
> >
> > [2] Cubuk et al. (2019)  AutoAugment
> >
> > [3] Lim et al (2019) Fast AutoAugment
> >
> > [4] Cubuk et al. (2020)  RandAugment
> >
> > [5] Hataya et al. (2020) Faster AutoAugment
> >
> > [6] Li et al. (2020) DADA
> >
> > [7] Rommel et al. (2022) CADDA

---

### Official Review · Reviewer_Km4K · 2022-07-11

**Rating:** 5
**Confidence:** 4
**Soundness:** 2 fair
**Presentation:** 3 good
**Contribution:** 2 fair

**Summary:**

The paper proposes a method to learn data invariance along with model training. The method avoids architectural modification and bilevel optimization, which makes it easy to use in many scenarios.

**Questions:**

My primary concern is the claim that the method can recover the *true* data invariance. It ideally would require two steps: (1) all possible types of invariance are considered in the augmentation module, and (2) the distribution is properly learned for each type of invariance. However, these two steps can hardly be satisfied. For the first step, (1a) it is impossible to enumerate all types of invariance; (1b) not all types of invariance are differentiable (e.g., cutoff); (1c) the paper does not discuss how to select the invariances among all possible combinations (e.g., using validation). For the second step, (2a) the scalar amplitude itself is insufficient to characterize an unparameterized distribution, and (2b) there is no theory in the paper that the recovered distribution matches the true distribution (Indeed, the learned distribution degenerates to identical mapping without regularizer). In summary, I can hardly agree that the learned augmentation matches the *true* invariance.

It also makes the motivation of the paper unclear. One reason to use augmentation is to boost performance --- however, the proposed method still falls behind the fixed augmentation. Another reason to use augmentation is to boost robustness against invariance attack --- however, such robustness is not systematically evaluated in the paper.

My last minor concern regards computational complexity. As mentioned in the paper, the model in inference requires more than one sample per example. I wonder if the authors could provide an analysis of the tradeoff between computational complexity and model accuracy/uncertainty.

**Limitations:**

Not applicable.

**Strengths And Weaknesses:**

The technical details are generally good and easy to follow; however, I am concerned about the motivation and theoretical foundation of the paper.

---

> ### Author Response · Authors · 2022-08-02
> **Detailed answer**
>
> We thank the reviewer for their remarks pointing to some clarifications regarding our actual contribution as well as suggesting stronger experimental results on CIFAR10. We hope that the answers below address these concerns as well as the **novel and improved experiments in CIFAR10**.
>
>
> Specifically, the reviewer asks clarifications concerning three main points:
> A) the basis on which we claim that we learn ‘’true invariances’’,
> B) the paper motivation as a data augmentation method and its performance compared to fixed augmentations on CIFAR10
> C) the trade-off between computational complexity and accuracy of our method.
>
> Point A)
> ---------
> Concerning the first point A), the term ‘’true invariance’’ is only used on line 4 of the abstract and line 136 of section 3.3, and it refers to the invariances of the underlying data distribution. The notion of invariance is briefly defined on line 110: we say a function $f$ is invariant to a set of transformations $G$ if for any $g$ in $G$, $f(gx)=f(x)$.
>
> The first concern (1a) raised by the reviewer here is that to learn any arbitrary true invariance one would have to enumerate all possible transformations of the data that exist. But **we never claim to learn all data invariances** and only **assume that the data is invariant to at least one transformation from our discrete pool $\mathcal{T}$**. This is not a strong assumption and we consider here augmentations that were proved useful in some contexts. This is now made clearer in section 3.3. Furthermore, we demonstrate on synthetic examples 4.1 and 4.2 **where we know at least one invariance of the data** that our model is able to learn it without any problem, which is one of the main points of these experiments. The number of invariances AugNet can learn depends indeed on the number of layers used and on the set of possible transformations $\mathcal{T}$. Joining what is asked by reviewer keKu, the fact that we cannot learn invariances to any arbitrary endomorphism $T:\mathcal{X} \to \mathcal{X}$ and can only select transformations from a finite pool is indeed a limitation of AugNet but is not specific to our work. It is the setting used by the whole field of automatic data augmentation ([1, 2, 3, 4, 5, 6]). This limitation and a discussion have been added to the end of the manuscript.

---

> > ### Author Response · Authors · 2022-08-02
> > **Detailed answer (2/3)**
> >
> > Still concerning point A), the reviewer also raised the concern (1b) that not all existing augmentations can be learned since some are not differentiable, giving the example of Cutout. As briefly mentioned in line 72 of our related work section, there are standard techniques to relax most non-differentiable augmentations into differentiable surrogates, as first explained in Faster AutoAugment [4] and reused in many following gradient-based automatic data augmentation works [5, 7]. We use these same relaxations in our work, **which is explained in lines 146-149 of section 3.3., as well as in section A.4 of supplementary.** Using these techniques, Cutout can be relaxed and made differentiable using for example the straight-through gradient estimator [8]. Also, as a case in point, **cutout augmentations on the time and sensors dimensions (time masking and channels dropout resp.) were learned with gradient descent in our EEG experiment of section 5.2** (see table A.4).  A discussion about this has been added to our new limitations and discussion section.
> >
> > We are not sure to understand the concern (1c) regarding the selection of transformations combinations. If by ‘’combination’’ the reviewer means the order in which the transformations should be applied, our method allows to learn this. Indeed, by stacking two or more augmentation layers, they will each select a different transformation at each layer, as shown on figures B.2-3 and B.5-6. An explanation of why each layer selects a single transformation is given in section 4.4.
> >
> > Lastly, the reviewer raises a concern about learning the ‘’true distribution’’ of invariances (2a and 2b). **We believe that there is a misunderstanding regarding this point**, since we never speak about or define a ‘’**true** distribution of invariances’’. As explained in section 3.1, we only define invariances of the model $f$ regarding **sets $G$ of deterministic transformations** (c.f. definition recalled in point above). In this paper we are trying to learn the set $G$ (which is potentially continuous), using a discrete set of typical augmentations as a **tool**. These augmentations are stochastic mappings, which hence represent **user-defined parametric distributions over many transformations**, since they have hyperparameters allowing to set how strongly they can distort the input data at the boundary of the distribution support. In practice, we use uniform distributions whose bounds are parameters to be learned, which allow us to create models $\tilde{f}$ approximately invariant to the subset $G$ defined by the union of the supports of the selected augmentations (c.f. proposition 3.1). So concerning point (2) in general, there is no such thing as ‘’true distribution of invariances’’ to be learned, and the distributions over transformations are just tools used in our model to approximate an invariant expectation.

---

> > > ### Author Response · Authors · 2022-08-02
> > > **Detailed answer (3/3)**
> > >
> > > Point B)
> > > ---------
> > >
> > > The reviewer found the motivation of our new data augmentation approach unclear because we don’t improve over all baselines. To clarify, our motivation is threefold:
> > > 1. Allow to learn useful data invariances and incorporate them in predictive models **without the need for manual search**,
> > > 2. Simplify automatic data augmentation with a new gradient-based method trainable end-to-end, without the need for complex bilevel optimization,
> > > 3. Extend previous work which allows to learn invariances end-to-end [8] with a more modular architecture, capable of learning a broader range of transformations (beyond the Lie group of affine symmetries). This last point will allow fields like medical applications of AI to have access to these techniques which are mostly reserved for computer vision.
> > >
> > > The reviewer mentions the results of the CIFAR10 experiment of section 5.1, where we cannot beat the fixed augmentations baseline. First of all, the gap between AugNet and the fixed augmentation baseline has been significantly reduced by increasing the number of copies $C$ in AugNet, as explained in our reply to reviewer GmuX. The two methods lead now to equivalent performance (AugNet: 93.2 +- 0.4, Fixed augmentation: 93.6 +- 0.2).
> > >
> > > Furthermore, note that while our augmentation technique does not lead to superior accuracy than this baseline, it allows to obtain the same level of performance without any prior knowledge of what augmentation will work. Indeed, it has been shown since AutoAugment [1] that the best augmentation to use highly depends on the dataset, model and task being solved. Hence, **consider the case where we are not working on CIFAR10**, for which we know which augmentations work best, **but are rather working with data from a real-world application for which augmentations have not been explored as extensively** as for computer vision benchmarking datasets (like sleep stage classification for example). In these cases, we usually have augmentation candidates (because it is a known modality for which augmentations have been proposed in the literature), but **we have no idea of which one will work best given the data, model and task being solved**. Our results on sections 5.1 and 5.2 show that our method allows to train a model end-to-end and obtain a performance comparable to what we would get after trying many augmentations combinations manually. It also shows now that it can outperform AutoAugment trained for thousands of hours, as well as methods based on  complex bilevel optimization with gradient-based automatic data augmentation methods such as Faster AutoAugment [4] or DADA [5]. In that sense, experiment 5.1 on CIFAR10 demonstrates that AugNet is actually a very competitive method (motivation 1), while experiment 5.2 on EEG MASS dataset makes a clear case that AugNet is a promising technique beyond computer vision problems (motivations 2 and 3). This has been made clearer in sections 5.1.
> > >
> > > Concerning robustness to adversarial attacks: this is out of the scope of our study, but could be an interesting direction for future work.
> > >
> > > Point C:
> > > ---------
> > >
> > > We thank the reviewer for suggesting this experiment, which indeed strengthens the analysis of our approach. We have included a graph comparing inference times and accuracies with varying C in section B.2. We observe that larger values of $C$ yield better performances.
> > > However, increasing the number of copies $C$ at inference also comes with a computation time that increases linearly.
> > >
> > >
> > >
> > > [1] Cubuk et al. (2019)  AutoAugment
> > >
> > > [2] Lim et al (2019) Fast AutoAugment
> > >
> > > [3] Cubuk et al. (2020)  RandAugment
> > >
> > > [4] Hataya et al. (2020) Faster AutoAugment
> > >
> > > [5] Li et al. (2020) DADA
> > >
> > > [6] Ho et al. (2019) Population Based Augmentation
> > >
> > > [7] Rommel et al. (2022) CADDA
> > >
> > > [8] Benton et al. (2020) Learning Invariances in Neural Networks

---

### Author Response · Authors · 2022-08-02
**General rebuttal summary**

We would like to thank the reviewers for their remarks and suggestions. Please note that the manuscript was updated with new figures and changes highlighted. We have answered in details to each reviewer in separate replies, but are summarizing here the main points of our rebuttal to facilitate the work of the AC:

- Reviewer 1 (Km4K) mainly criticizes the fact that we cannot learn all invariances of a dataset, nor its ‘’true distributions of invariance’’. We believe there is a misunderstanding since **we don’t do these claims in the paper** (cf answer to reviewer for more details).
- Reviewers 1 (Km4K) and 3 (kEKu) criticize the fact that our method **requires a pool $\mathcal{T}$ of candidate augmentations**, but this is not specific to our work and is the **general framework of nearly all automatic data augmentation papers** [1-8].
- Reviewers 2 (7J9r) and 3 (kEKu) challenge the **novelty of AugNet compared to Augerino** [1]. To summarize:
1. Augerino can only sample affine transformations while **we can sample any differentiable augmentation**;
2. Hence, **Augerino’s scope is limited to applications such as computer vision**, while AugNet has a large scope;
3. Thanks to our new regularizer and new modular architecture in layers, AugNet can learn **sequences of augmentations** in an optimal order (like employed in [2;3;6]), which was **not possible with Augerino**;
4. Lastly, we have shown in our CIFAR10 experiment of section 5.1 that **AugNet outperforms Augerino**.
- **Reviewer 3 (kEKu) seems to have missed the point 2 above**, which is why they criticized us for not comparing to Augerino in experiments which are out of its scope (sinusoids and EEG experiments of sections 4.2, 4.3 and 5.2).
- Following reviewers 1 (Km4K), 3 (kEKu) and 4 (GmuX) suggestions, we have now **significantly improved our experiments** (fig 6a). We have indeed **increased AugNet’s accuracy** on our CIFAR10 experiment by simply increasing the number of augmentations sampled $C$ and have **added two new strong baselines** to it (AutoAugment [2] and RandAugment [4]), both outperformed by AugNet. We have also **added four new sensitivity analyses on AugNet’s hyperparameters** $C$ and $\lambda$ and extended our experiment on model invariance to the larger CIFAR10 dataset (section B.2). The latter clarifies why it is ok to have magnitudes dropping to 0 (reviewer 4 GmuX).
- Concerning the extension of our experimental results in general, our manuscript now presents **10 different experiments** (2 benchmarks, 3 qualitative analysis of learned invariance, 1 ablation study, 3 hyperparameter sensitivity analysis, 1 invariance vs model capacity analysis) in a total of **4 datasets** (2 synthetic and 2 real) with **2 different data types** (images and signals), **23 different augmentation operations**, and a total of **7 strong SOTA baselines**. We don’t focus on very large computer vision datasets such as ImageNet (reviewer 3 and 4) because we believe automatic data augmentation is all-the-more important for applications where data is scarce (e.g. neuroscience and other medical applications), which is the scope of this work.
- Furthermore, as advised by most of the reviewers, **we have added a new section discussing the main limitations of our work** (need for a pool of candidate augmentations, limitation to augmentations amenable to differentiable relaxation and AugNet’s trade-off between performance and inference time).

[1] Benton et al. (2020) Learning Invariances in Neural Networks

[2] Cubuk et al. (2019)  AutoAugment

[3] Lim et al (2019) Fast AutoAugment

[4] Cubuk et al. (2020)  RandAugment

[5] Hataya et al. (2020) Faster AutoAugment
[6] Li et al. (2020) DADA
[7] Ho et al. (2019) Population Based Augmentation
[8] Rommel et al. (2022) CADDA

---

### Meta-Review · Area_Chair_yfxv · 2022-08-25

**Recommendation:** Accept
**Confidence:** Less certain

**Metareview:**

The decision for this paper was a hard one. I pondered the scores with respect to the engagement of the different reviewers.  I believe the initial scores were due to a misunderstanding of the limitations of the baseline model Augerino, and how the proposed method solves some of the failures and limitations of Augerino (e.g. being able to model only affine transformation). I also find the authors expanded their experiments in a convincing manner during the rebuttal period. We encourage the authors to *improve the clarity of their contributions* in their final version, and to include all additional experiments that were ran during the rebuttal period.


**Award:**

No

---

### Decision · Program_Chairs · 2022-09-14

Accept